# Micro and Macro Level Graph Modeling for Graph Variational Auto-Encoders

**Kiarash Zahirnia, Oliver Schulte,* Parmis Naddaf, Ke Li**
School of Computing Science, Simon Fraser University, Canada
{kzahirni, oschulte, pnaddaf, keli}@sfu.ca

## Abstract

Generative models for graph data are an important research topic in machine learning. Graph data comprise two levels that are typically analyzed separately: node-level properties such as the existence of a link between a pair of nodes, and global aggregate graph-level statistics, such as motif counts. This paper proposes a new multi-level framework that *jointly* models node-level properties and graph-level statistics, as mutually reinforcing sources of information. We introduce a new *micro-macro* training objective for graph generation that combines node-level and graph-level losses. We utilize the micro-macro objective to improve graph generation with a GraphVAE, a well-established model based on graph-level latent variables, that provides fast training and generation time for medium-sized graphs. Our experiments show that adding micro-macro modeling to the GraphVAE model improves graph quality scores up to 2 orders of magnitude on five benchmark datasets, while maintaining the GraphVAE generation speed advantage.

## 1 Introduction: multi-level graph modeling

Many datasets contain relational information about entities and their links that can be represented as a graph. Deep generative learning on graphs has become a popular research topic [20], with applications including molecule design [42], and recommendation [12]. It is common in graph analysis to distinguish two levels of information [10, 20]: (1) local node-level properties, such as the existence of a link between two nodes or the attribute of a node [27, 37, 19], and (2) global graph-level statistics (graph statistics for short) that depend on the entire graph, such as node degree distribution or motif counts. Most deep Graph Generative Models (GGMs) are trained with an objective based on local properties (e.g., maximizing the probabilities of observed edges). Figure 1 illustrates the difference between local and global structure. Different edges have different roles in the graph global structure. Some edges play a critical role for the connectivity/community structure, while others are less important. Previous GGM training is generally based on a likelihood objective that decomposes into individual edge likelihoods [47, 31, 11], which does not discriminate edges with different roles. Graph global level information on the other hand can capture how different edges have different importance in the graph structure (see also Section 4). Whereas the sparsity of typical graphs causes difficulties for training objectives based on local information only [38, 27], graph statistics are typically dense.

This paper proposes a new perspective on the node/graph level dichotomy: a principled probabilistic framework that incorporates both local and global graph properties. From the framework we derive a novel micro-macro (MM) objective function for training a GGM to match *both* local and global properties. The term *micro-macro* originates from the philosophy of science and refers to scientific frameworks that treat local ("micro") resp. global ("macro") properties of a system as equal targets [17].

---

*Supported by NSERC Canada Discovery Grant R611341

36th Conference on Neural Information Processing Systems (NeurIPS 2022).

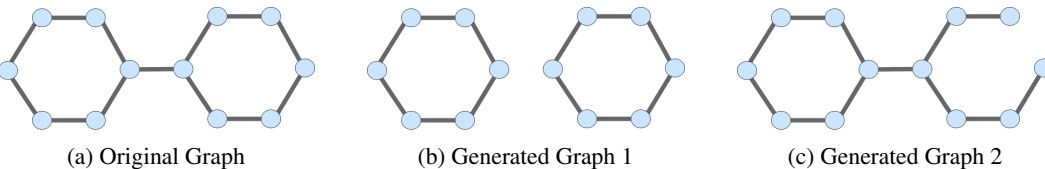

| (a) Original Graph | (b) Generated Graph 1 | (c) Generated Graph 2 |

Figure 1: To illustrate the difference between local and global properties. The two right graphs (1b), (1c) score the same in terms of number of edges reconstructed from the left graph (1a), a local node-level property. However the right graph (1c), is structurally more similar to (1a), containing the same number of connected components, a global graph-level statistic.

**Advantages** Micro-macro modeling increases *graph realism* and *user control*. (1) Compared to objective functions that are based on predicting local properties, matching graph statistics serves as a *regularizer* that increases the realism of the generated graph structures. This is sufficient to generate realistic graphs with a fast all-at-once edge-parallel model. (2) O'Bray et al. [35] note that different graph statistics are important for different applications. For example for a large payment graph recording economic transactions, a macro economist may be mainly interested in the average price level of a goods basket. For a central bank managing a payment system, the total number of transactions may be more important. O'Bray et al. [35] therefore advise selecting a GGM based on the graph statistics of interest in the target application. However, this entails searching through the space of GGMs and their hyperparameters to find a good match with target graph statistics. In our MM framework, the user only needs to specify the target graph statistics and learning will automatically select graph models that match them.

**Approach** We introduce a joint probabilistic model over both local properties and global graph statistics. An MM objective that can be used with graph encoder-decoder models is derived as an ELBO from the joint probability model. Our experiments focus on the GraphVAE (Graph Variational Auto-Encoder) model where the encoder outputs a graph-level latent posterior $z$ (graph embedding) and the decoder maps the graph embedding $z$ to a soft adjacency matrix representing link probabilities [43]. A graph-level embedding supports modeling graph-level statistics. We show how the recent calibrated Gaussian variational auto-encoder [41] can be adapted to model graph statistics with divergent scales. The implementation and datasets are provided at `https://github.com/kiarashza/GraphVAE-MM`.

**Evaluation** Evaluating GGMs has become a research topic of its own [44, 35]; see the related work, section below. Proposed metrics quantify how similar a set of generated graphs is to a set of observed graphs. Our assessment focuses on evaluation metrics based on Graph Neural Networks (GNN-based metrics) [44]. To our knowledge, this is the most recent published evaluation method, with state-of-the-art (SOTA) performance shown in extensive experiments. For GraphVAEs, adding micro-macro modeling improves both the realism and the diversity of the generated graphs, up to an order of magnitude on GNN-based metrics. This is sufficient to reach very competitive graph quality compared to SOTA auto-regressive benchmark models [32, 47, 11]. At the same time, one-shot graph generation with GraphVAEs is faster than sampling from auto-regressive models. Our experiments on medium-sized graphs show that even with the overhead of micro-macro modeling, GraphVAEs maintain their generation speed advantage over auto-regressive approaches. While our evaluation focuses on GraphVAEs, we discuss how micro-macro modeling can be applied to other GGMs to leverage graph-level statistics.

*Contributions.* Our main contributions can be summarized as follows.

- A new joint probabilistic model over both local graph properties and global graph-level statistics.

- Deriving a joint ELBO as a new micro-macro objective function for training graph encoder-decoder models.

- An adaptation of the GraphVAE architecture [43] to learn graph embeddings using the joint micro-macro objective.

## 2  Related work

*Graph Statistics as Modeling Targets.* To our knowledge, this is the first work to develop a GGM with the objective of matching graph statistics. Work on graph moment matching in network statistics [36, 6] has studied theoretical properties of graph moment estimators with increasing graph size. Maximizing the likelihood of exponential random graph models [21] such as Markov Logic Networks is equivalent to maximizing entropy subject to matching a set of graph statistics [28].

*Graph Statistics for Evaluating Generative Models.* The ability of generative models to match an empirical graph statistic distribution has been assessed in several research papers, which supports our approach of including them in the training objective. Farnadi et al. [16] compare relational models in terms of matching (expected) motif counts. You et al. [47] and Liao et al. [32] use 4 graph statistics to measure the realism of generated graphs.

The question of how to evaluate a GGM has been studied in recent papers [35, 44]. An evaluation metric quantifies how similar a set of generated graphs $\hat{G}_1, \ldots, \hat{G}_{m_1}$ is to a set of test graphs $G_1, \ldots, G_{m_2}$. From the graphs we can compute corresponding descriptors (e.g., statistics) $\hat{x}_1, \ldots, \hat{x}_{m_1}$ and $x_1, \ldots, x_{m_2}$. The similarity of two descriptor sets can be quantified using Maximum Mean Distance (MMD). O'Bray et al. [35] raise several difficulties with this approach. (1) Given a list of graph statistics, we obtain a list of MMD scores, rather than a single score that can be used to rank GGMs. (2) The statistics-based MMD scores are sensitive to hyper-parameters; different settings can lead to different model rankings even for a single graph statistic. (3) In perturbation studies where observed graphs are corrupted in a controlled manner, MMD scores do not correlate well with degree of perturbation. Thompson et al. [44] develop a recent proposal to address these issues. They utilize a *reference embedding network* GNN $\mathcal{E}$. The embedder $\mathcal{E}$ is obtained from pretraining or from random weights and is therefore independent of any of the models under evaluation. Given $m_1$ generated graphs and $m_2$ test graphs, $\mathcal{E}$ provides embeddings $\hat{e}_1, \ldots, \hat{e}_{m_1}$ and $e_1, \ldots, e_{m_2}$ which can be compared using a vector metric such as MMD. Extensive evaluation shows that the GGM scores provided by the reference embedding approach correlate well with perturbation degree and capture the realism and diversity of generated graphs. To our knowledge, the GNN-based approach is the state of the art for evaluating GGMs, so *we use it as our main evaluation metric.*

*Graph Generation Architecture.* Our novel contribution is to develop a new kind of objective function, not a new kind of GGM. We therefore follow an AB design where we utilize an existing architecture as is and change only the training objective. Developing GGMs is an on-going topic of research; overviews can be found in [50, 20]. Two major GGM groups are all-at-once and auto-regressive methods [20]. All-at-once generation decodes a latent variable $\boldsymbol{z}$ to generate a (soft) adjacency matrix $\tilde{\boldsymbol{A}}$. Auto-regressive methods [32, 47, 25] generate a graph incrementally. All-at-once methods are faster at graph generation, but tend to generate less realistic graphs. They can use node-level representations or a graph-level representation (embedding). We use a graph-level representation for two reasons: (1) A natural fit with modeling graph-level statistics. (2) It is known from previous work that node-level representations, while useful for many applications, do not capture enough graph structure to generate realistic graphs. We utilize the well-known GraphVAE architecture for computing graph embeddings [43],[20, Ch.9 Sec.1.2].

For a generative model of graph statistics, we adapt calibrated Gaussian variational auto-encoder [41], which has been developed as a generative model for i.i.d. data, but not previously been applied to graph modeling.

## 3  Data model and micro-macro objective

An attributed graph is a pair $G = (V, E)$ comprising a finite set of $N$ nodes and edges where each node is assigned an $d$-dimensional attribute $\boldsymbol{X}_i$. An attributed graph can be represented by an $N \times N$ adjacency matrix $\boldsymbol{A}$ with $\{0, 1\}$ entries, together with an $N \times d$ node feature matrix $\boldsymbol{X}$. Following Ma et al. [33], we view the observed adjacency matrix as a sample from an underlying probabilistic adjacency matrix $\tilde{\boldsymbol{A}}$ with $\tilde{\boldsymbol{A}}_{i,j} \in [0, 1]$. The sampling distribution for independent edges is given by

$$p(\boldsymbol{A}|\tilde{\boldsymbol{A}}) = \prod_{i=1}^{N}\prod_{j=1}^{N} \tilde{\boldsymbol{A}}_{i,j}^{\boldsymbol{A}_{i,j}}(1 - \tilde{\boldsymbol{A}}_{i,j})^{1-\boldsymbol{A}_{i,j}}. \tag{1}$$

A **descriptor function** $\phi$ maps an adjacency matrix $\boldsymbol{A}$ to a $l$-dimensional **graph statistic** such that $\phi(\boldsymbol{A}) \in \mathbb{R}^l$ [35]. We use also matrices as graph statistics. We consider only descriptor functions that are permutation-equivariant [20, Ch.5]. A graph statistic represents a higher-order graph property. Examples include a node degree histogram or motif counts; for discussion and further examples see [20, Sec.2.1.2] and Section 4 below.

### 3.1 Micro-macro objective

We assume a finite list of descriptor functions $\phi_1, \ldots, \phi_m$ that define **target statistics**.

For a fixed attribute matrix $\boldsymbol{X}$, a **micro-macro (MM) loss** is of the form

$$\mathcal{L}_{\boldsymbol{\theta}}(\boldsymbol{A}) = \mathcal{L}_{\boldsymbol{\theta}}^0(\boldsymbol{A}) + \gamma \mathcal{L}_{\boldsymbol{\theta}}^1(\boldsymbol{F}_1, \ldots, \boldsymbol{F}_m)$$

where $\boldsymbol{A}$ is the training graph and $\boldsymbol{F}_u, u = 1, \ldots, m$ is a random variable defined by applying the $u$-th descriptor function to the training graph. The hyperparameter $\gamma$ controls the balance between capturing micro and macro aspects of the training graph. The micro loss $\mathcal{L}^0$ decomposes into losses for each node or pair of nodes. Well-known examples include the cross-entropy loss and max-margin loss. In this paper we work with negative log-likelihood losses and a variational approximation based on an encoder-decoder architecture:

$$\mathcal{L}_{\boldsymbol{\psi}}^0(\boldsymbol{A}) = -\ln p_{\boldsymbol{\psi}}^0(\boldsymbol{A}) = -\ln \int p(\boldsymbol{A}|\tilde{\boldsymbol{A}}_{\boldsymbol{z}})p(\boldsymbol{z})d\boldsymbol{z} \tag{2}$$

$$\mathcal{L}_{\boldsymbol{\psi},\boldsymbol{\sigma}}^1(\boldsymbol{F}_1, \ldots, \boldsymbol{F}_m) = -\sum_{u=1}^{m} \frac{1}{|\boldsymbol{F}_u|} \ln p_{\boldsymbol{\psi},\boldsymbol{\sigma}}^1(\boldsymbol{F}_u) \tag{3}$$

$$p_{\boldsymbol{\psi},\boldsymbol{\sigma}}^1(\boldsymbol{F}_u) = \int \mathcal{N}(\boldsymbol{F}_u|\phi_u(\tilde{\boldsymbol{A}}_{\boldsymbol{z}}), \sigma_u^2 I)p(\boldsymbol{z})d\boldsymbol{z} \tag{4}$$

using the following notation.

- $\boldsymbol{z}_{1 \times t}$ specifies a latent $t$-dimensional graph embedding with prior distribution $p(\boldsymbol{z})$.

- $\tilde{\boldsymbol{A}}_{\boldsymbol{z}}$ is a probabilistic adjacency matrix computed as a trainable deterministic decoder function of graph embedding $\boldsymbol{z}$. The decoder parameters are denoted as $\boldsymbol{\psi}$. The edge reconstruction probability $p(\boldsymbol{A}|\tilde{\boldsymbol{A}}_{\boldsymbol{z}})$ is computed as in Equation (1).

- The conditional distribution of each graph statistic is modeled as a Gaussian $\mathcal{N}$ with diagonal variance parameter $\sigma_u^2$:

$$p_{\boldsymbol{\psi},\boldsymbol{\sigma}}^1(\boldsymbol{F}_u|\boldsymbol{z}) = \mathcal{N}(\boldsymbol{F}_u|\phi_u(\tilde{\boldsymbol{A}}_{\boldsymbol{z}}), \sigma_u^2 I).$$

  The Gaussian mean is computed by applying the $u$-th descriptor function to the reconstructed (soft) adjacency matrix. Even with a Gaussian conditional distribution, the *marginal* distribution over graph statistics, $p_{\boldsymbol{\psi},\boldsymbol{\sigma}}^1(\boldsymbol{F}_u)$, can in principle fit any distribution [26].

- $|\boldsymbol{F}_u|$ is the dimensionality of target statistic $\boldsymbol{F}_u$.

Figure 2a shows a generative model diagram for Equations (2)–(4). Normalizing each graph statistic by its dimension is important because their scales can diverge widely. For example, transition matrices have $N^2$ entries, while the number of triangles is a single scalar. The following proposition provides an ELBO for an MM loss.

**Proposition 1** *Let $\mathcal{L}_{\boldsymbol{\theta}}(\boldsymbol{A})$ be a micro-macro loss defined by Equations* (2)*–* (4)*. Then*

$$\mathcal{L}_{\boldsymbol{\theta}}(\boldsymbol{A}) \leq E_{\boldsymbol{z} \sim q_{\varphi}(\boldsymbol{z}|\boldsymbol{A},\boldsymbol{X})}\Big[ -\ln p(\boldsymbol{A}|\tilde{\boldsymbol{A}}_{\boldsymbol{z}}) - \gamma \sum_{u=1}^{m} \big( \frac{1}{|\boldsymbol{F}_u|} \ln \mathcal{N}(\boldsymbol{F}_u|\phi_u(\tilde{\boldsymbol{A}}_{\boldsymbol{z}}), \sigma_u^2 I)\Big] \tag{5}$$
$$+ (1 + \gamma m)KL(q_{\varphi}(\boldsymbol{z}|\boldsymbol{A},\boldsymbol{X})||p(\boldsymbol{z}))$$

*where $q_{\varphi}(\boldsymbol{z}|\boldsymbol{A},\boldsymbol{X}) = q(\boldsymbol{z}|\boldsymbol{A},\boldsymbol{X},\boldsymbol{F}_1, \ldots, \boldsymbol{F}_m)$ is an approximate posterior distribution with parameters $\varphi$.*

The proof is in the Appendix section 8.1. The basic idea is to combine ELBOs for each individual loss term. Adding a hyperparameter $\beta$ to control the relative importance of the KL-divergence (as in the $\beta$-VAE [22]), we obtain the following *training objective* for a set of observed training graphs $\boldsymbol{A}^1, \ldots, \boldsymbol{A}^n$:

$$\underset{\varphi, \psi, \sigma}{\arg\min} \sum_{j=1}^{n} E_{\boldsymbol{z} \sim q_\varphi(\boldsymbol{z}|\boldsymbol{A}^j, \boldsymbol{X})} \big[ -\ln p(\boldsymbol{A}^j|\tilde{\boldsymbol{A}}_{\boldsymbol{z}}) - \gamma \sum_{u=1}^{m} \frac{1}{|\boldsymbol{F}_u|} \ln \mathcal{N}(\boldsymbol{F}_u^j | \phi_u(\tilde{\boldsymbol{A}}_{\boldsymbol{z}}), \sigma_u^2 I) \big] \qquad (6)$$
$$+ \beta KL(q_\varphi(\boldsymbol{z}|\boldsymbol{A}^j, \boldsymbol{X}) \| p(\boldsymbol{z}))$$

In this objective, *the graph statistic reconstruction loss*, $-\gamma \sum_{u=1}^{m} \frac{1}{|\boldsymbol{F}_u|} \ln \mathcal{N}(\boldsymbol{F}_u^j | \phi_u(\tilde{\boldsymbol{A}}_{\boldsymbol{z}}), \sigma_u^2 I)$, *acts as a regularizer with respect to the edge reconstruction loss* $-\ln p(\boldsymbol{A}^j|\tilde{\boldsymbol{A}}_{\boldsymbol{z}})$.

### 3.2 Implementation

We implement the micro-macro ELBO (6) utilizing graph neural networks as follows.

- The prior $p(\boldsymbol{z})$ is a standard normal distribution.
- We limit the encoder input to the observed training graph, so $q(\boldsymbol{z}|\boldsymbol{A}, \boldsymbol{X}, \boldsymbol{F}_1, \ldots, \boldsymbol{F}_m) = q_\varphi(\boldsymbol{z}|\boldsymbol{A}, \boldsymbol{X})$. Therefore we can use any standard graph encoder with parameters $\varphi$. The encoder deterministically maps an input graph $(\boldsymbol{A}, \boldsymbol{X})$ to a posterior distribution $q_\varphi(\boldsymbol{z}|\boldsymbol{A}, \boldsymbol{X})$.
- The decoder deterministically maps a latent representation $\boldsymbol{z}$ to a probabilistic graph $\tilde{\boldsymbol{A}}_{\boldsymbol{z}}$ with parameters $\psi$. Any standard graph decoder can be used.
- For each graph-level statistic, the diagonal co-variance parameter is computed using the optimal $\sigma$-VAE method from the calibrated Gaussian framework [41]. This uses the maximum likelihood estimate given the estimated means with respect to a minibatch of size $B$:

$$\sigma_u^2 = \frac{1}{B} \sum_{j=1}^{B} MSE(\phi_u(\tilde{\boldsymbol{A}}_{\boldsymbol{z}^j}), \boldsymbol{F}_u^j)), \qquad MSE(\boldsymbol{x}, \boldsymbol{\mu}) = \frac{1}{|\mu|} \sum_{l=1}^{|\mu|} (\boldsymbol{x}_l - \boldsymbol{\mu}_l)^2.$$

Figure 2b plots the architecture. The advantages of the calibrated Gaussian model for generating graph statistics are as follows. (1) In experiments on non-relational data, the calibrated Gaussian model achieves state-of-the-art performance. (2) Learning a variance parameter eliminates hyperparameters compared to user-assigned weights for each graph statistic. (3) The variance values can be interpreted as quantifying the empirical uncertainty of a graph feature. For example, if all training graphs exhibit a similar number of triangles, the corresponding variance parameter will be small. Table 6 in the Appendix illustrates this phenomenon in benchmark datasets.

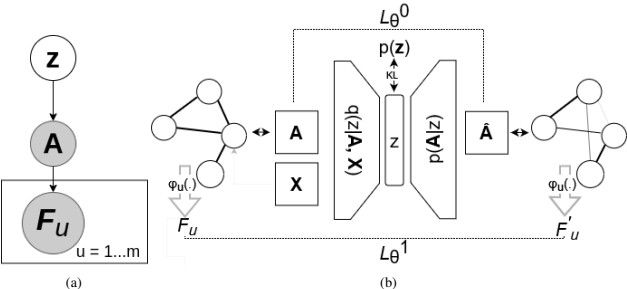

Figure 2: (a) Micro-macro generative model. Observed variables in gray. (b) GraphVAE–MM architecture.

## 4 Graph statistics

The micro-macro ELBO (6) requires user-specified input graph descriptors for computing target graph statistics. In our experiments, we utilize several types of *default statistics* for regularizing

graph embeddings that have the following advantages. (1) Known from prior research to be generally important for graph modeling across different domains. (2) Easy to interpret. (3) Differentiable with respect to the entries in a reconstructed soft adjacency matrix. In an application, the default statistics can be combined with other statistics of interest.

Graph descriptors used in network analysis and graph kernels comprise two main groups [20]: (1) Summaries of local node-level properties. We utilize a descriptor derived from the degree histogram, and the number of triangles in a graph.(2) Higher-order proximity relations between nodes. We utilize $s$-step transition probability matrices, for $s = 1, \ldots 5$ [9, 49]. These descriptors are differentiable with respect to the entries in a reconstructed soft adjacency matrix $\tilde{A}$, which we view as representing a weighted undirected graph.

**Degree Histogram.** The **degree** of node $v_i$ is given by $d(v_i) = \sum_j \tilde{A}_{ij}$. We adapt the permutation-invariant differentiable histogram layer (DHL) [45].

The DHL is based on a soft assignment of points to bins given bin centers and widths. Binning the soft degrees forms a soft degree histogram of graph $\tilde{A}$ comparable to the hard degree histogram. In detail, the bin centers are the possible (hard) node degrees $b = 0, \ldots, N$. All widths are uniformly set to 0.1 (based on experimentation). Then we have

$$\mathbf{d}_{\tilde{A}}(b) \equiv \sum_{i=1}^{N} a(v_i, b), \qquad a(v_i, b) = \max\{0, 1 - 0.1 \cdot |d(v_i) - b|\}.$$

Thus the membership $a(v_i, b)$ of a node in a bin $b$ ranges from 0 to 1 and decreases with the difference between the node's expected degree and the bin center. The DHL assigns to each bin the sum of nodes membership in the bin. In a naive implementation, the computational cost of finding the $\mathbf{d}_{\tilde{A}}(b)$ vector is $O(N^2)$. In our experiments, we use $N$ parallel processors for a near constant time to obtain the soft degree histogram.

$S$**-Step Transition Probability Kernel.** $P^s(\tilde{A})$ is the $N \times N$ $s$-step transition probability matrix derived from adjacency matrix $\tilde{A}$ such that $[P^s(\tilde{A})]_{i,j}$ is the probability of a transition from node $i$ to node $j$ in a random walk of $s$ steps started from $i$. The transition matrix $P^s(\tilde{A})$ can be computed as $P^s(\tilde{A}) = (D(\tilde{A})^{-1}\tilde{A})^s$ where $D(\tilde{A})$ is a diagonal matrix with $D(\tilde{A})_{ii} = d(v_i)$.

The GraRep system learns node representations that reconstruct the $s$-step transition probabilities, using random walks and matrix factorization [9]. The transition probability matrix is usually dense and encodes the connectivity information of the graph. An important difference to the adjacency matrix is that whereas adding or removing an edge changes only one adjacency, it can and often does result in a substantive change in many transition matrix elements. The descriptor $P^s(\tilde{A})$ thus measures the importance of an edge in the graph structure (cf. Figure 1). In our experiments, we compute the transition probability matrix exactly with a runtime cost of $O(N^3)$.

**Triangle Count.** The number of triangles in a simple graph $A$ is computed by $Tri(A) = \sum_i (A^3)_{ii}$

with a runtime cost of $O(N^3)$. The number of triangles is a fundamental graph statistic in network science [5] with many applications in graph mining [1]. For example it has been used to detect spamming activity and assess content quality in social networks [4], to uncover thematic structure in the world-wide web [14] and for query planning in databases [3]. It is also used extensively in exponential random graph models [21].

## 5 Empirical evaluation

We describe our baseline methods and benchmark datasets, then report comparison results.

### 5.1 Comparison methods

We examine the effect of micro-macro modeling on a GraphVAE architecture. Our AB methodology is to keep the architecture the same and train the model using the joint MM ELBO (6) that combines both local and global graph properties. We also compare the MM GraphVAE with popular GGMs. As

the architectures are not new, we describe them briefly. Our experiments used the public repositories and recommended hyperparameters by original creators; the Appendix contains further details.

**GraphVAE.** A popular model that transforms a graph-level latent variable to generate a soft adjacency matrix [43][20, Ch.9.1.2]. The encoder utilizes a Multi-layer Graph Convolutional Network (GCN) followed by a graph level readout and a Fully Connected Layer (FCL) that outputs $(\mu, \sigma)$. The decoder utilizes a network with FCLs that outputs entries of a probabilistic adjacency matrix $\tilde{A}$. To evaluate the reconstruction probability, we use a BFS ordering of nodes [31][20, Ch.9.1.2].
**GraphVAE–MM.** GraphVAE trained with the micro-macro objective 6. **Our new method.**
**GraphRNN.** Auto-regressive method that generates the adjacency matrix incrementally. Each step generates one entry in the GraphRNN design, or one column in the GraphRNN-S design [47].
**GRAN.** Auto-regressive method that generates a block of nodes and associated edges at a step [32].
**BiGG.** Auto-regressive method that leverages graphs sparsity to avoid generating the full adjacency matrix [11]. To our knowledge BiGG achieves SOTA graph quality.

## 5.2 Benchmark datasets

Our design closely follows previous experiments on generating realistic graph structures [47, 32]. We utilize 3 synthetic, and 2 real-world datasets for the main results. The synthetic Grid and Lobster are from previous studies [47, 32], Triangle Grid is introduced in this paper. Protein and ogbg-molbbbp are real-world datasets from biology with information about proteins and molecules respectively. The Appendix, section 8.2, contains further details, as well as results for 3 more real-world datasets.

*Train/Test Split.* Following previous experiments [47, 30, 11] we randomly split the graphs sets into train (70%), validation (10%) and test (20%) sets. We use the same train and test graph sets for all models. To evaluate a trained model, we generate $T$ new graphs to compare with the $T$ graphs in the test set (cf. Section 2). Appendix Figure 5 illustrates the evaluation process.

Network modeling at the micro-level might be used for surveillance of individuals and communities. Therefore ethical network data collection must take into account issues of consent, privacy, and bias. None of the datasets used in this research study contain personally identifiable information, or offensive/harmful content about either individuals or communities. The social impact of our work we expect to be on balance more positive than negative, because our macro-level model enhances the understanding of global network structure, not the targeting of individuals.

## 5.3 Evaluation

We empirically verify the effectiveness of micro-macro modeling through different evaluation metrics: Qualitative and quantitative evaluation of graph quality, generation time, and training time.

**Qualitative Evaluation.** Following [47, 32] we compare the generated graphs by visual inspection. Figure 3 provides a visual comparison of randomly selected test graphs and generated graphs. GraphVAE-MM and BiGG achieve the best visual match. The Appendix, section 8.5, provides more examples.

**Quantitative Evaluation.** We follow the descriptor-based approach established in previous works (cf. Section 2). As recommended by O'Bray et al. [35], we include scores computed from a 50/50 split of the data set as an ideal score, i.e., a lower bound on the GGM scores.

(1) The GNN-based evaluation metrics [44] MMD RBF and F1 PR compare generated and test graph embeddings computed by a reference GNN with randomly initialized weights. The reference embeddings are independent of any target graph statistics used in GraphVAE-MM. F1 PR stands for "Improved Precision and Recall". Precision is the percentage of generated graph embeddings that fall within the manifold of test graph embeddings, while recall is the percentage of test graph embeddings that fall within the manifold of generated graph embeddings. This metric mainly captures the *diversity* of generated graphs [35]. MMD RBF compares generated graph embeddings and test graph embeddings using an RBF kernel. This metric captures the *realism* of the generated graphs.

(2) A statistics-based evaluation compares generated and test graphs using *evaluation statistics*, namely node degree, clustering coefficient, orbit counts, and the spectra of the graphs [32, 47]. We add graph diameter, a statistic commonly studied in network science [29] related to graph connectivity. The target statistics used to train a GraphVAE-MM model are distinct from the evaluation statistics

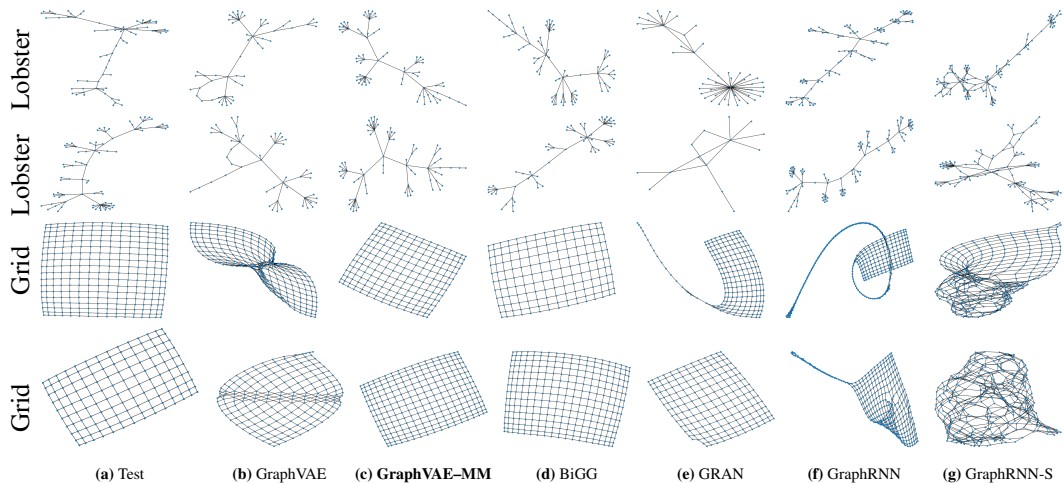

Figure 3: Visualization of generated graphs. The left-most column (3a) shows randomly selected graphs from the test set for each dataset, the other columns show graphs generated by each model from the prior. The generated graphs shown are the two visually most similar samples in the generated set. GraphVAE–MM achieves the best visual match, much better than GraphVAE.

used to measure graph quality (however, node degree and Degree histogram are closely related). For each statistic, MMD is computed using total variation (TV) distance [32].

**Impact on GraphVAE.** Table 1 shows a very large improvement from micro-macro modeling for both MMD RBF and F1 PR. The improvement in F1 PR ranges from 6%-40%, achieving a near perfect score. For MMD RBF the reduction ranges from 0.06 to 0.26 which is significant given the low ideal score. Table 2 shows the results for evaluation graph statistics. Micro-macro modeling reduces MMD by 1-2 orders of magnitude on almost all datasets. In sum, MM modeling provides a large improvement in the realism and diversity of graphs generated by a GraphVAE architecture.

Lesion studies in the Appendix investigate the effects of each target statistic in isolation (Table 5). No single statistic has the power of all three combined. We also observe that different target statistics have different importance for different datasets; see Table 6.

Table 1: GNN-based evaluation of micro-macro modeling for GraphVAEs. For each dataset we report the MMD RBF and F1 PR (see text) between the test graphs (left-most column in Figure 3) and the generated graphs (see other columns in Figure 3). Values reported are the mean $\pm$ std. For MMD RBF smaller values are better, for F1 PR larger values are better.

| Method | Triangle Grid | | Lobster | | Grid | | ogbg-molbbbp | | Protein | |
|---|---|---|---|---|---|---|---|---|---|---|
| | MMD RBF | F1 PR | MMD RBF | F1 PR | MMD RBF | F1 PR | MMD RBF | F1 PR | MMD RBF | F1 PR |
| 50/50 split | $0.03 \pm 0.00$ | $98.99 \pm 0.00$ | $0.04 \pm 0.00$ | $98.58 \pm 0.00$ | $0.009 \pm 0.00$ | $98.70 \pm 0.00$ | $0.002 \pm 0.00$ | $98.07 \pm 0.00$ | $0.04 \pm 0.00$ | $98.67 \pm 1.11$ |
| GraphVAE | $0.23 \pm 0.01$ | $75.92 \pm 8.96$ | $0.36 \pm 0.11$ | $78.48 \pm 24.13$ | $0.17 \pm 0.01$ | $75.52 \pm 2.53$ | $0.20 \pm 0.07$ | $54.53 \pm 6.15$ | $0.10 \pm 0.05$ | $84.11 \pm 9.56$ |
| GraphVAE-MM | $\mathbf{0.17 \pm 0.01}$ | $\mathbf{83.58 \pm 5.50}$ | $\mathbf{0.10 \pm 0.00}$ | $\mathbf{100.00 \pm 0.00}$ | $\mathbf{0.13 \pm 0.01}$ | $\mathbf{97.09 \pm 6.33}$ | $\mathbf{0.02 \pm 0.01}$ | $\mathbf{93.78 \pm 1.33}$ | $\mathbf{0.03 \pm 0.01}$ | $\mathbf{90.78 \pm 3.76}$ |

**GraphVAE-MM vs. Benchmark GGMs.** For benchmarking we include GGMs that are known to generate realistic graphs. Tables 3 shows the GNN-based quality scores. Other than the most recent BiGG method, GraphVAE-MM achieves a much better score. MMD RBF scores, and 3 out of 5 F1 PR scores are also better in GraphVAE-MM compared to BiGG. Triangle Grid shows the biggest improvement, which illustrates the usefulness of matching triangle counts for this dataset.

Appendix Table 7 shows the benchmark results of statistics-based evaluation; we summarize them here. On synthetic graphs, the GraphVAE-MM scores are superior to or competitive with the BiGG and GRAN scores. On the real-world graphs, the GraphVAE-MM scores are competitive with the BiGG and GRAN scores, and superior to the other benchmarks. Given the already strong performance of the auto-regressive models, we conclude that GraphVAE-MM generates high-quality graphs.

**Generation Time.** The code for all models is run on the same system, detailed in Appendix section 8.11. Figure 4 compares GraphVAEs to the fastest auto-regressive methods. The auto-regressive methods require substantially more generation time than GraphVAEs. While MM modeling slows

Table 2: Statistics-based evaluation of micro-macro modeling for GraphVAEs. For a named evaluation graph statistic, each column reports the MMD between the test graphs and the generated graphs.

(a) Synthetic Graphs

| Method | Triangle Grid | | | | | Lobster | | | | | Grid | | | | |
|---|---|---|---|---|---|---|---|---|---|---|---|---|---|---|---|
| | Deg. | Clus. | Orbit | Spect | Diam. | Deg. | Clus. | Orbit | Spect | Diam. | Deg. | Clus. | Orbit | Spect | Diam. |
| 50/50 split | $3e^{-5}$ | 0.002 | $8e^{-5}$ | 0.004 | 0.014 | 0.002 | 0 | 0.002 | 0.005 | 0.032 | $1e^{-5}$ | 0 | $2e^{-5}$ | 0.004 | 0.014 |
| GraphVAE | 0.082 | 0.442 | 0.421 | 0.020 | 0.152 | 0.081 | 0.739 | 0.372 | 0.056 | **0.129** | 0.062 | 0.055 | 0.515 | 0.018 | 0.143 |
| GraphVAE-MM | **0.001** | **0.093** | **0.001** | **0.013** | **0.133** | **$2e^{-4}$** | **0** | **0.008** | **0.017** | 0.187 | **$5e^{-4}$** | **0** | **0.001** | **0.014** | **0.065** |

(b) Real Graphs

| Method | Protein | | | | | ogbg-molbbbp | | | | |
|---|---|---|---|---|---|---|---|---|---|---|
| | Deg. | Clus. | Orbit. | Spect. | Diam. | Deg. | Clus. | Orbit. | Spect. | Diam. |
| 50/50 split | $4e^{-5}$ | 0.004 | $5e^{-4}$ | $4e^{-4}$ | 0.003 | $2e^{-4}$ | $2e^{-5}$ | $9e^{-5}$ | $5e^{-4}$ | 0.002 |
| GraphVAE | 0.022 | 0.108 | 0.577 | 0.016 | **0.080** | 0.028 | 0.442 | 0.047 | 0.015 | 0.055 |
| GraphVAE-MM | **0.006** | **0.059** | **0.152** | **0.007** | 0.091 | **0.001** | **0.005** | **$8e^{-4}$** | **0.005** | **0.018** |

Table 3: GNN-based comparison with benchmark GGMs. See table 1 caption. The best result is in bold and the second best is underlined.

| Method | Triangle Grid | | Lobster | | Grid | | ogbg-molbbbp | | Protein | |
|---|---|---|---|---|---|---|---|---|---|---|
| | MMD RBF | F1 PR | MMD RBF | F1 PR | MMD RBF | F1 PR | MMD RBF | F1 PR | MMD RBF | F1 PR |
| 50/50 split | $0.03 \pm 0.00$ | $98.58 \pm 0.00$ | $0.04 \pm 0.00$ | $98.58 \pm 0.00$ | $0.009 \pm 0.00$ | $98.70 \pm 0.00$ | $0.002 \pm 0.00$ | $98.07 \pm 0.00$ | $0.04 \pm 0.00$ | $98.67 \pm 1.11$ |
| GraphVAE-MM | $\mathbf{0.17 \pm 0.01}$ | $83.58 \pm 5.50$ | $\mathbf{0.10 \pm 0.00}$ | $\mathbf{100.00 \pm 0.00}$ | $\mathbf{0.13 \pm 0.01}$ | $97.09 \pm 6.33$ | $0.02 \pm 0.01$ | $93.78 \pm 1.33$ | $\mathbf{0.03 \pm 0.01}$ | $90.78 \pm 3.76$ |
| GraphRNN-S [47] | $0.72 \pm 0.17$ | $33.68 \pm 19.44$ | $0.98 \pm 0.13$ | $58.72 \pm 7.55$ | $0.79 \pm 0.08$ | $71.18 \pm 2.36$ | $0.48 \pm 0.02$ | $81.41 \pm 0.71$ | $0.28 \pm 0.26$ | $72.36 \pm 27.63$ |
| GraphRNN [47] | $0.64 \pm 0.11$ | $25.80 \pm 11.75$ | $0.87 \pm 0.04$ | $61.97 \pm 6.04$ | $0.99 \pm 0.03$ | $13.22 \pm 2.05$ | $1.45 \pm 0.19$ | $\mathbf{98.94 \pm 0.56}$ | $0.32 \pm 0.14$ | $93.94 \pm 0.56$ |
| GRAN [32] | $0.88 \pm 0.09$ | $23.71 \pm 9.72$ | $0.24 \pm 0.04$ | $50.53 \pm 12.12$ | $0.40 \pm 0.00$ | $78.73 \pm 0.02$ | $0.39 \pm 0.07$ | $94.06 \pm 2.60$ | $0.07 \pm 0.00$ | $98.05 \pm 0.76$ |
| BiGG [11] | $0.41 \pm 0.13$ | $62.08 \pm 0.14$ | $0.12 \pm 0.00$ | $99.74 \pm 0.76$ | $0.35 \pm 0.00$ | $92.43 \pm 0.00$ | $0.04 \pm 0.00$ | $96.16 \pm 0.31$ | $0.15 \pm 0.00$ | $\mathbf{98.11 \pm 0.62}$ |

down training, for the GraphVAE, the training time is still less than for the auto-regressive methods. The Appendix table 8 provides time measurements for all methods. The Appendix Table 12 presents worst-case complexity bounds of the comparison methods.

## 5.4 Further experiments on real-world graphs

We conducted further experiments on the MUTAG, PTC [46], and QM9 [43] datasets to evaluate micro-macro modeling on more real world graphs. The detailed results are in the Appendix section 8.10; we summarize them here. For all datasets, GraphVAE-MM offers much faster generation than the auto-regressive baselines. On MUTAG and PTC, the improvements from micro-macro modeling are even better than those we reported on the Protein and ogbg-molbbbp datasets. GraphVAE-MM achieves substantive improvement in generation quality over all baselines, except for BiGG on PTC, which is competitive. It was infeasible to train the auto-regressive methods on the QM9 datasets (except for BiGG), so we report only the comparison of GraphVAE vs. GraphVAE-MM. Micro-macro modeling brings small improvement in graph quality on QM9, which is good given the strong score of GraphVAE on QM9.

## 6 Limitations and discussion

We discuss the limitations of our GraphVAE-MM model and options for combining micro-macro modeling with other GGMs.

*Strengths and Weaknesses.* GraphVAE-MM inherits the strengths of GraphVAE [20, Ch.9.1.2]: expressive power through graph embeddings, and fast generation due to all-at-once parallel edge generation. GraphVAE–MM also inherits the known limitations of GraphVAEs: 1) We need to know a maximum number of nodes before generation. (Smaller graphs can be generated using a mask.) 2) The decoder is an FCL that ouputs $N \times N$ numbers; we need to (implicitly) assume a node ordering to evaluate edge reconstruction probabilities based on the FCL output. The dependence on a node ordering is common to both GraphVAEs and auto-regressive GGMs, and efficient heuristics have been designed whose effectiveness has been confirmed in experiments, including those reported in this paper. We note that MM modeling makes GGM training less dependent on a node ordering because it uses permutation-invariant graph statistics.

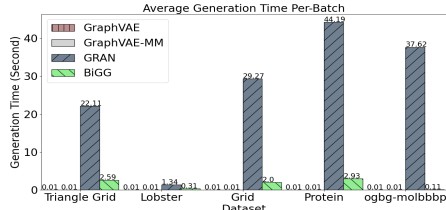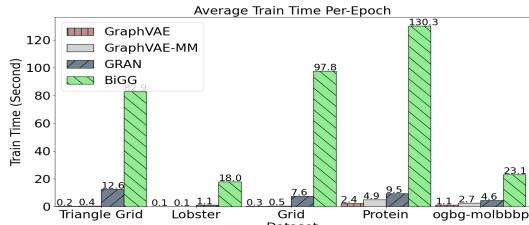

Figure 4: Comparison of benchmark GGMs with GraphVAEs on *generation time* (left) and *training time* (right). For visualizing the small generation time of GraphVAE and GraphVAE–MM , we round them up to 0.01, the first and second bar of each dataset respectively.

*Training Time Overhead.* Evaluating the GraphVAE-MM ELBO during training incurs computational overhead compared to the GraphVAE ELBO due to computing graph statistics. We note that evaluating the edge reconstruction probability is already expensive and can dominate training time. Table 12 in the Appendix presents worst-case complexity bounds for the training time of our comparison methods, which indicate that scaling GraphVAEs to very large graphs is a challenge. Despite the theoretical worst-case cost of evaluating graph statistics, Figure 4(right) shows that the overhead was small in our experiments with medium-size graphs, due to parallelization. Also, approximating graph statistics is a promising avenue for significantly reducing the training time even for large graphs significantly [24, 15, 31, 40].

*Micro-Macro Modeling for Other GGM Architectures.* In principle the MM objective can be applied with auto-regressive architectures as well: after a complete graph has been generated sequentially, the global loss can be backpropagated through the individual edge generation decisions.

In a graph GAN [7, 8] the generator maps a graph latent $z$ to an adjacency matrix, or a random walk and the discriminator classifies them as real or synthetic. GAN models are not considered as SOTA GGMs [11]. Both random walk and adjacency matrices represent local micro-level information only [9]. A straightforward way to combine MM modeling with GANs is to augment the input to the discriminator with graph statistics computed for both real and generated graphs.

# 7    Conclusion and future work

Our main idea is to model graph data jointly at two levels: a micro level based on local information (e.g. the existence of a link between two nodes) and a macro level based on aggregate graph statistics. We described a principled joint probabilistic model for both micro and macro levels, and derived an ELBO training objective for graph encoder-decoder models. Compared to previous micro level training objectives, the macro statistics regularize graph embeddings to match global graph statistics. To evaluate our model, we described a set of strong default graph statistics (node degree, number of triangles, transition probabilities). We applied the new training objective to the GraphVAE architecture, a widely used graph generative model based on latent graph representation. Micro-macro (MM) modeling greatly improved the quality of graphs generated by GraphVAE, to match or exceed that of benchmark models. MM modeling maintains the speed advantages of edge-parallel all-at-once graph generation. With an efficient computation of graph statistics, it provided fast training time as well compared to auto-regressive methods.

Micro-macro modeling opens a number of fruitful avenues for future work. i) Investigating which graph statistics are important for generating which types of graphs. This connects with the rich area of graph kernels [34] that are often based on graph statistics. ii) Investigating which graph statistics are important for particular domains. iii) Developing a micro-macro model for other graph generative architectures, such as auto-regressive and GANs. iv) Evaluating the impact of micro-macro modeling on other graph-level tasks, such as graph classification.

In sum, modeling both global graph statistics and local information enhances the power of graph generation. Compared to using local information only, graph statistics can be used to regularize graph representations to efficiently generate realistic graphs.

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
