# 8 Appendix

## 8.1 Proof of proposition 1

$$\mathcal{L}_{\boldsymbol{\theta}}(\boldsymbol{A}) = \mathcal{L}_{\boldsymbol{\theta}}^0(\boldsymbol{A}) + \gamma\mathcal{L}_{\boldsymbol{\theta}}^1(\boldsymbol{F}_1,\ldots,\boldsymbol{F}_m)$$

$$\leq E_{\boldsymbol{z}\sim q_{\varphi}(\boldsymbol{z}|\boldsymbol{A},\boldsymbol{X},\boldsymbol{F}_1,\ldots,\boldsymbol{F}_m)}\big[-\ln p(\boldsymbol{A}|\tilde{\boldsymbol{A}}_{\boldsymbol{z}})\big] + KL(q_{\varphi}(\boldsymbol{z}|\boldsymbol{A},\boldsymbol{X},\boldsymbol{F}_1,\ldots,\boldsymbol{F}_m)||p(\boldsymbol{z})) \tag{7}$$

$$-\gamma\sum_{u=1}^m\frac{1}{|\boldsymbol{F}_u|}\big(E_{\boldsymbol{z}\sim q_{\varphi}(\boldsymbol{z}|\boldsymbol{A},\boldsymbol{X},\boldsymbol{F}_1,\ldots,\boldsymbol{F}_m)}[\ln\mathcal{N}(\boldsymbol{F}_u|\phi_u(\tilde{\boldsymbol{A}}_{\boldsymbol{z}}),\sigma_u^2 I]$$

$$+ KL(q_{\varphi}(\boldsymbol{z}|\boldsymbol{A},\boldsymbol{X},\boldsymbol{F}_1,\ldots,\boldsymbol{F}_m)||p(\boldsymbol{z})))$$

$$= E_{\boldsymbol{z}\sim q_{\varphi}(\boldsymbol{z}|\boldsymbol{A},\boldsymbol{X},\boldsymbol{F}_1,\ldots,\boldsymbol{F}_m)}\big[-\ln p(\boldsymbol{A}|\tilde{\boldsymbol{A}}_{\boldsymbol{z}}) - \gamma\sum_{u=1}^m\frac{1}{|\boldsymbol{F}_u|}\ln\mathcal{N}(\boldsymbol{F}_u|\phi_u(\tilde{\boldsymbol{A}}_{\boldsymbol{z}}),\sigma_u^2 I)\big] \tag{8}$$

$$+ KL(q_{\varphi}(\boldsymbol{z}|\boldsymbol{A},\boldsymbol{X},\boldsymbol{F}_1,\ldots,\boldsymbol{F}_m)||p(\boldsymbol{z})) + \gamma\sum_{u=1}^m\frac{1}{|\boldsymbol{F}_u|}\big([KL(q_{\varphi}(\boldsymbol{z}|\boldsymbol{A},\boldsymbol{X},\boldsymbol{F}_1,\ldots,\boldsymbol{F}_m)||p(\boldsymbol{z})))$$

$$\leq E_{\boldsymbol{z}\sim q_{\varphi}(\boldsymbol{z}|\boldsymbol{A},\boldsymbol{X},\boldsymbol{F}_1,\ldots,\boldsymbol{F}_m)}\big[-\ln p(\boldsymbol{A}|\tilde{\boldsymbol{A}}_{\boldsymbol{z}}) - \gamma\sum_{u=1}^m\frac{1}{|\boldsymbol{F}_u|}\ln\mathcal{N}(\boldsymbol{F}_u|\phi_u(\tilde{\boldsymbol{A}}_{\boldsymbol{z}}),\sigma_u^2 I)\big] \tag{9}$$

$$+ KL(q_{\varphi}(\boldsymbol{z}|\boldsymbol{A},\boldsymbol{X},\boldsymbol{F}_1,\ldots,\boldsymbol{F}_m)||p(\boldsymbol{z})) + \gamma\sum_{u=1}^m\big([KL(q_{\varphi}(\boldsymbol{z}|\boldsymbol{A},\boldsymbol{X},\boldsymbol{F}_1,\ldots,\boldsymbol{F}_m)||p(\boldsymbol{z})))$$

$$= E_{\boldsymbol{z}\sim q_{\varphi}(\boldsymbol{z}|\boldsymbol{A},\boldsymbol{X},\boldsymbol{F}_1,\ldots,\boldsymbol{F}_m)}\big[-\ln p(\boldsymbol{A}|\tilde{\boldsymbol{A}}_{\boldsymbol{z}}) - \gamma\sum_{u=1}^m\frac{1}{|\boldsymbol{F}_u|}\ln\mathcal{N}(\boldsymbol{F}_u|\phi_u(\tilde{\boldsymbol{A}}_{\boldsymbol{z}}),\sigma_u^2 I)\big] \tag{10}$$

$$+ (1+\gamma m)KL(q_{\varphi}(\boldsymbol{z}|\boldsymbol{A},\boldsymbol{X},\boldsymbol{F}_1,\ldots,\boldsymbol{F}_m)||p(\boldsymbol{z}))$$

Inequality (7) adds ELBOs for each individual loss term. Equation (8) uses the linearity of expectation. Inequality(9) follows because the fact that $|\boldsymbol{F}_u| \geq 1$. Equation (10) collects the KL expressions into a single term, and establishes the inequality (5).

## 8.2 Benchmark datasets

Following previous studies [47, 32], we utilize synthetic and real graph datasets as follows.

**Triangle Grid.** Includes 100 synthetic 2D graphs, regular tiling of the 2D plane with equilateral triangles, with $100 \leq |V| < 400$ [47].
**Lobster.** Includes 100 synthetic graphs with $10 \leq |V| \leq 100$. Generated using the code from [47].
**(Square) Grid.** Includes 100 synthetic 2D graphs, regular tiling of the 2D plane with equilateral squares, with $100 \leq |V| < 400$ [47].
**Protein.** Consists of 918 real-world protein graphs with $100 \leq |V| \leq 500$ [13].
**ogbg-molbbbp.** Consists of 2039 real-world molecular graphs with $2 \leq |V| \leq 132$ [23].

## 8.3 Flowchart for evaluating generated graphs

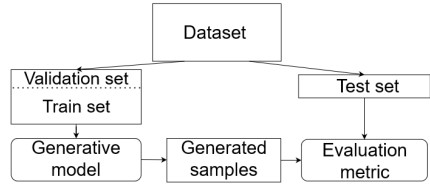

Figure 5: Each dataset is split into train, validation and test sets. A model is trained on the train set. The trained model generates new samples. The generated samples are then compared with the test set.

Figure 5 illustrates the evaluation approach. To split the datasets and for statistic-based evaluation metrics we use the code from [32]. For GNN-based metrics we used the code from [44]. Following

Goyal et al. [18], for all the GGMs if a graph with disconnected components is generated, we take the maximum connected component.

## 8.4 Setup and hyper-parameters

**GraphVAE.** We examine the effect of micro-macro modeling on the GraphVAE architecture [43]. GraphVAE **encoder** utilizes a two-layer GCN (256, 1026 dimension) followed by the graph-level output formulation (sum of nodes representation) finalized with FCL (1024 dimension) outputting the parameters of the variational posterior distribution. The model assumes isotropic and standard Gaussian distributions for the variational posterior and prior, respectively. The GraphVAE decoder is a three-layer fully connected neural network (1024,1024,1024 dimension) that directly maps the graph representation to a probabilistic adjacency matrix. Layers utilize Layer Normalization [2] and LeakyReLU activation function. Our AB methodology is to keep the architecture the same and train the model A) using GraphVAE ELBO i.e. $\mathcal{L}_{\theta}^{0}(\boldsymbol{A})$, and B) the joint MM ELBO (6) that combines both local and global graph properties. GraphVAE and GraphVAE-MM are trained using the Adam optimizer [26] with a learning rate of 0.0003 for 20,000 epochs except for Lobster and ogbg-molbbbp, smaller datasets, which are trained for 10,000 epochs. For the hyperparameters $\gamma$ and $\beta$ see table 4. Hyperparameters are selected by validation set performance.

**Baselines.** For all Baselines, we used the implementation and hyperparameters setting provided by the original papers.

Table 4: $\gamma$ and $\beta$ hyperparameters for each dataset used in graph generation task.

| Dataset | Triangle Grid | Lobster | Grid | ogbg-molbbbp | Protein | MUTAG | PTC | QM9 |
|---|---|---|---|---|---|---|---|---|
| $\gamma$ | 50 | 40 | 50 | 40 | 50 | 4 | 2 | 80 |
| $\beta$ | $2e^3$ | $1.5e^3$ | $2e^3$ | $1.5e^3$ | $1e^3$ | 60 | 60 | 200 |

## 8.5 Qualitative evaluation of GGMs in detail

Here we extend the visual examination of the micro-macro modeling and benchmark GGMs from Figure 3. The AB methodology for evaluating the micro-macro modeling is to use the same VGAE architecture, here GraphVAE section 8.4, and applying micro-macro modeling. Figures 8, 7, 6, 10 and 9 provide visual comparisons of benchmark GGMs and the effect of the micro-macro modeling approach. For each model, we generate 20 samples and visually select and plot the most similar ones to the test set. The first block in each of the figures shows randomly selected target graphs from the test set. The second block compares the effect of the micro-macro modeling on the GraphVAE and the last block samples graphs generated by the benchmark GGMs.

## 8.6 Lesion studies on MM objective components

This section drills downs into the different components of the MM objective and the importance of KL-divergence penalty. The KL-divergence importance is studied by training the GraphVAE with $\beta$-VAE and different values for hyperparameter $\beta$ which controls the relative importance of the KL-divergence penalty. The result shows that while the KL-divergence weighting affects the model performance, it cannot replace macro modeling. See the first block of table 5. The second block of table 5 investigates the effects of each target statistic in isolation. As shown, no single statistic has the power of all three combined. We also observe that different statistics are more important for different datasets.

## 8.7 Variance parameter

Table 6 shows the values of the learned variance parameter $\sigma_u^2$ by the optimal $\sigma$-VAE approach for each of the datasets. The variance of a graph statistic can be interpreted as quantifying the empirical uncertainty of a graph statistic. Taking for example the *Triangle count* statistic, the learned variance for the Lobster, Grid, and ogbg-molbbbp datasets is very small because there are almost no triangles in these datasets. For the protein dataset, the learned variance is comparatively large, which indicates the number of triangles has a wider range of values in this dataset.

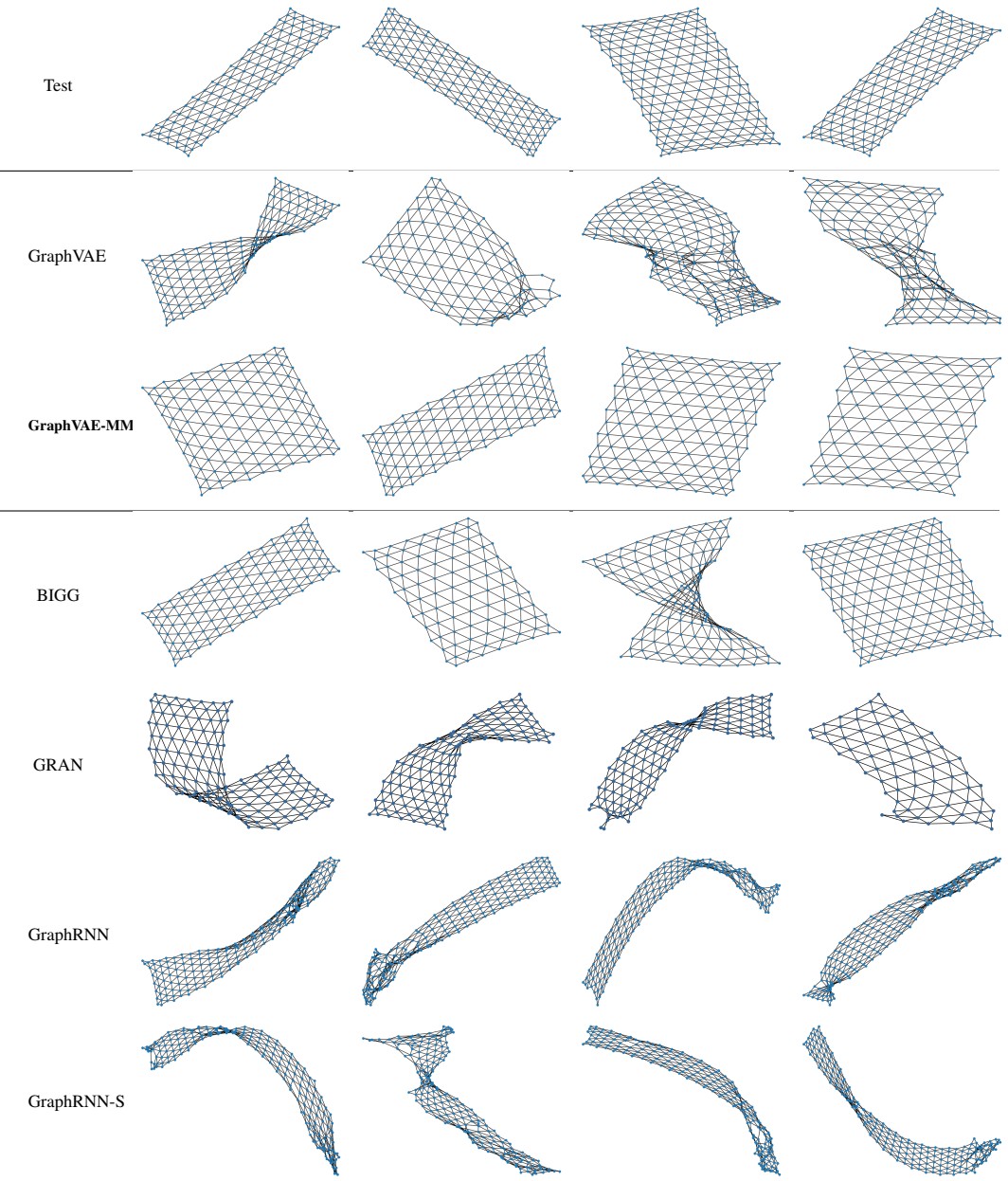

Figure 6: Visualization of generated **Triangle Grid** graphs by benchmark GGMs and the micro-macro modeling effect. The first block shows four randomly selected graphs from the test set. The first and second rows in the second block show samples generated by GraphVAE and GraphVAE-MM models respectively. The bottom block shows graphs generated by benchmark GGMs. Graphs generated with micro-macro modeling, GraphVAE-MM, match the target graph the best and make a noticeable improvement in comparison to GraphVAE. For each model we visually select and visualize the most similar generated samples to the test set.

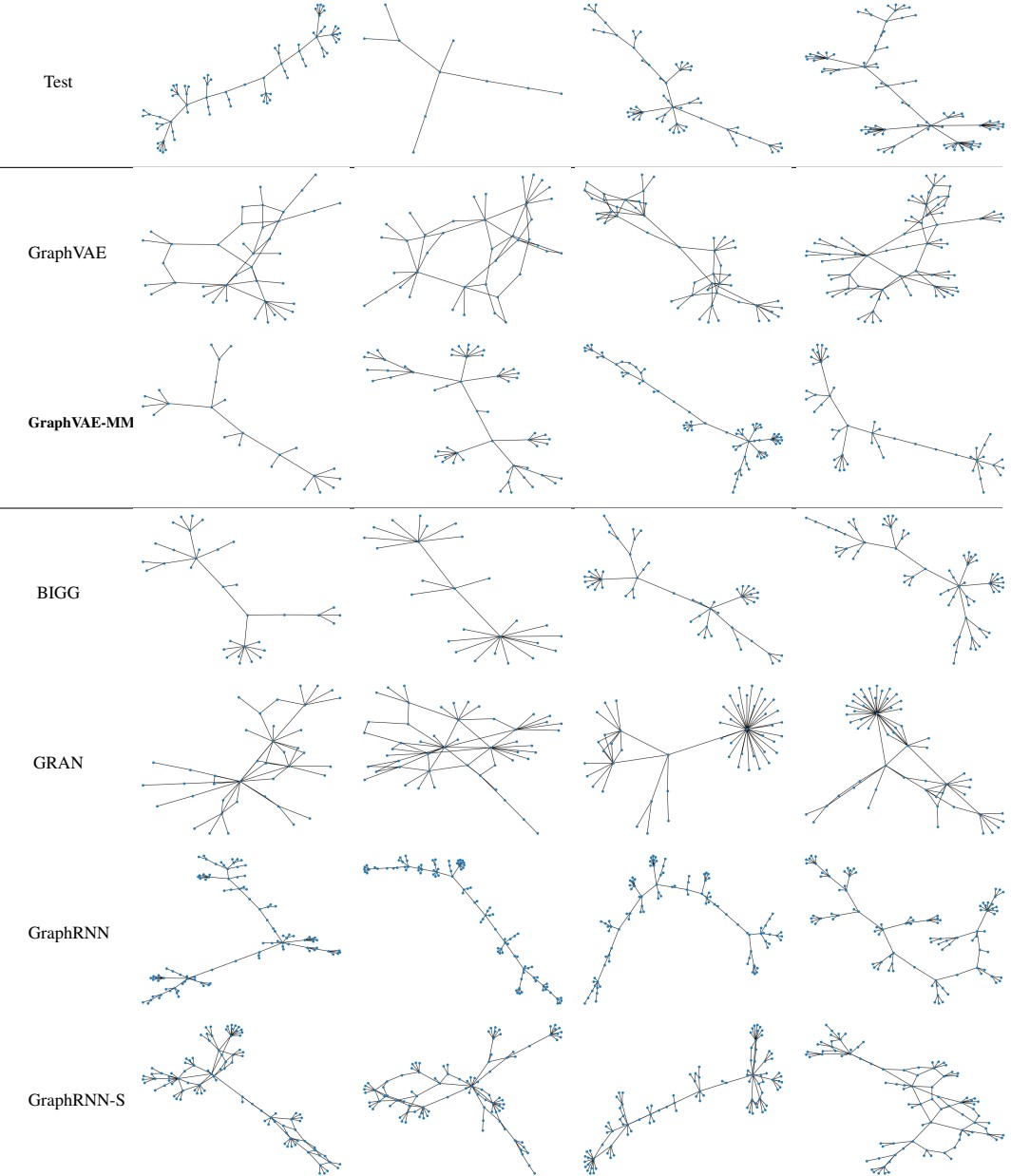

Figure 7: Visualization of generated **Lobster** graphs by benchmark GGMs and the micro-macro modeling effect. The first block shows four randomly selected graphs from the test set. The first and second rows in the second block show samples generated by GraphVAE and GraphVAE-MM models respectively. The bottom block shows graphs generated by benchmark GGMs. Graphs generated with micro-macro modeling, GraphVAE-MM, match the target graph the best and make a noticeable improvement in comparison to GraphVAE. For each model we visually select and visualize the most similar generated samples to the test set.

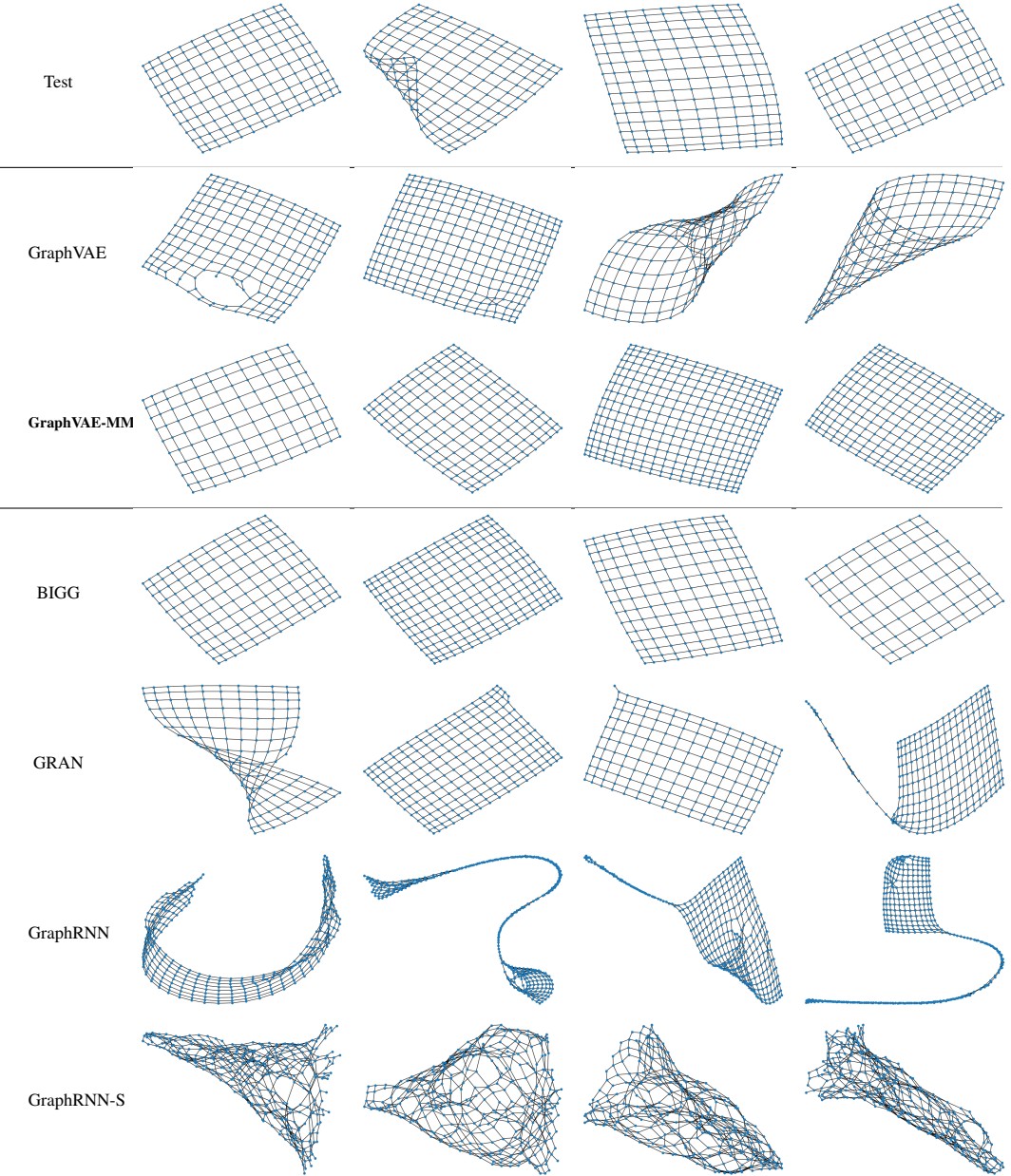

Figure 8: Visualization of generated **Grid** graphs by benchmark GGMs and the micro-macro modeling effect. The first block shows four randomly selected graphs from the test set. The first and second rows in the second block show samples generated by GraphVAE and GraphVAE-MM models respectively. The bottom block shows graphs generated by benchmark GGMs. Graphs generated with micro-macro modeling, GraphVAE-MM, match the target graph the best and make a noticeable improvement in comparison to GraphVAE. For each model we visually select and visualize the most similar generated samples to the test set.

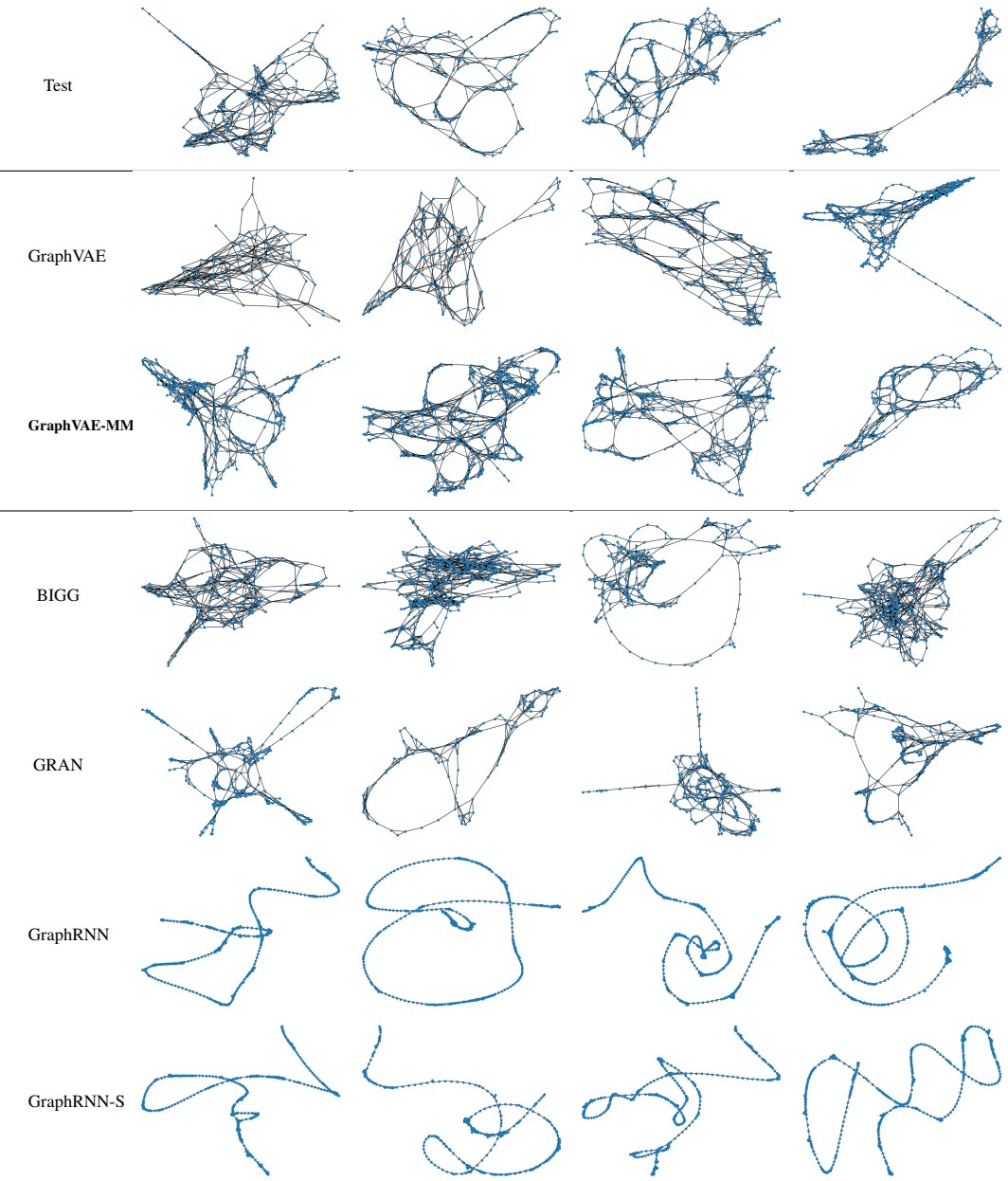

Figure 9: Visualization of generated **Protein** graphs by benchmark GGMs and the micro-macro modeling effect. The first block shows four randomly selected graphs from the test set. The first and second rows in the second block show samples generated by GraphVAE and GraphVAE-MM models respectively. The bottom block shows graphs generated by benchmark GGMs. Graphs generated with micro-macro modeling, GraphVAE-MM, make a noticeable improvement in comparison to GraphVAE. For each model we visually select and visualize the most similar generated samples to the test set.

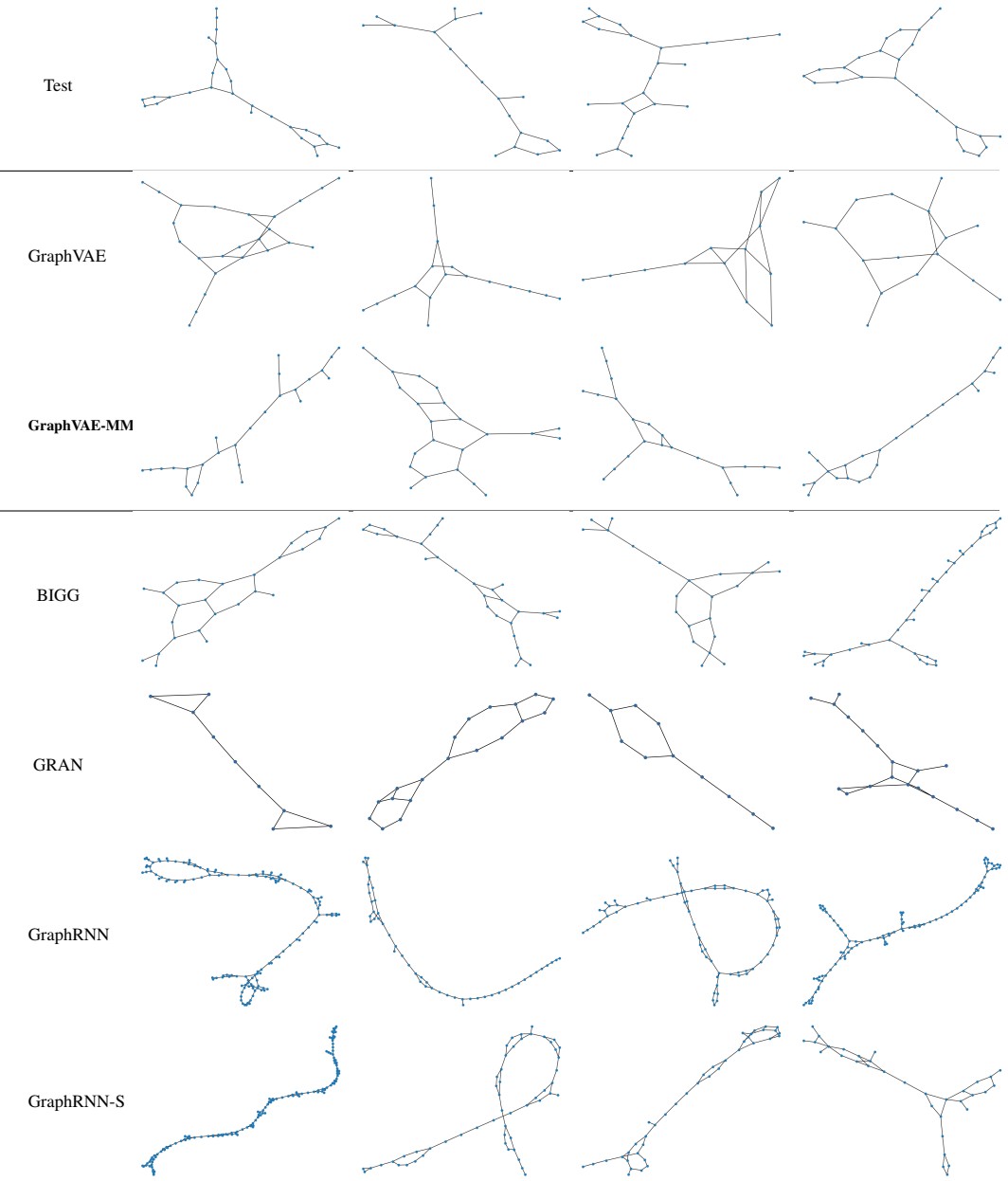

Figure 10: Visualization of generated **ogbg-molbbbp** graphs by benchmark GGMs and the micro-macro modeling effect. The first block shows four randomly selected graphs from the test set. The first and second rows in the second block show samples generated by GraphVAE and GraphVAE-MM models respectively. The bottom block shows graphs generated by benchmark GGMs. Graphs generated with micro-macro modeling, GraphVAE-MM, make a noticeable improvement in comparison to GraphVAE. For each model we visually select and visualize the most similar generated samples to the test set.

Table 5: Sensitivity Analysis and Lesion Studies on MM objective components. The first block shows the GraphVAE trained with $\beta$-VAE [22] and different $\beta$. $\beta$-VAE has a limited effect and cannot compensate for the lack of macro components. The second block investigates the effects of each target statistic in isolation. No single statistic has the power of all three combined.

| Method | Lobster | | | | | ogbg-molbbbp | | | | |
|---|---|---|---|---|---|---|---|---|---|---|
| | Deg. | Clus. | Orbit | Spect | Diam. | Deg. | Clus. | Orbit | Spect | Diam. |
| GraphVAE ($\beta = 1$) | 0.081 | 0.739 | 0.372 | 0.056 | 0.129 | 0.028 | 0.442 | 0.047 | 0.015 | 0.055 |
| GraphVAE ($\beta = 2$) | 0.039 | 0.324 | 0.262 | 0.031 | 0.021 | 0.022 | 0.425 | 0.032 | 0.015 | 0.028 |
| GraphVAE ($\beta = 4$) | 0.068 | 0.485 | 0.472 | 0.065 | 0.147 | 0.044 | 0.560 | 0.063 | 0.019 | 0.046 |
| GraphVAE ($\beta = 8$) | 0.080 | 0.432 | 0.465 | 0.065 | 0.085 | 0.127 | 0.736 | 0.250 | 0.034 | 0.055 |
| GraphVAE ($\beta = 16$) | 0.024 | 0.314 | 0.295 | 0.024 | **0.068** | 0.243 | 0.834 | 0.537 | 0.051 | 0.109 |
| GraphVAE ($\beta = 32$) | 0.099 | 0.513 | 0.594 | 0.045 | 0.148 | 0.392 | 0.975 | 0.896 | 0.103 | 0.266 |
| GraphVAE–MM | **$2e^{-4}$** | **0** | **0.008** | 0.017 | 0.187 | **0.001** | **0.005** | **$8e^{-4}$** | **0.005** | **0.018** |
| GraphVAE-Triangle-Count | 0.009 | **0** | 0.052 | **0.016** | 0.209 | 0.382 | 0.962 | 0.846 | 0.076 | 0.116 |
| GraphVAE-$s$-step | 0.015 | 0.255 | 0.092 | 0.025 | 0.097 | 0.399 | 0.943 | 0.866 | 0.097 | 0.336 |
| GraphVAE-Degree-Hist | 0.025 | 0.336 | 0.342 | 0.228 | 0.078 | 0.349 | 0.950 | 0.811 | 0.080 | 0.128 |

Table 6: Values of the learned variance parameter, $\sigma_u^2$, for each graph statistic. The variance values can be interpreted as quantifying the empirical uncertainty of a graph statistic.

| Dataset | 2-step | 3-Step | 4-step | 5-step | 6-step | Degree hist. | Triangle count |
|---|---|---|---|---|---|---|---|
| Triangle Grid | $3.93e^{-5}$ | $4.11e^{-5}$ | $4.39e^{-5}$ | $4.53e^{-5}$ | $4.63e^{-5}$ | 0.21 | 4.94 |
| Lobster | $1.18e^{-5}$ | $9.85e^{-6}$ | $9.61e^{-6}$ | $9.49e^{-6}$ | $9.45e^{-6}$ | $3.89e^{-5}$ | $6.87e^{-6}$ |
| Grid | $2.90e^{-5}$ | $2.16e^{-5}$ | $2.02e^{-5}$ | $1.92e^{-5}$ | $1.86e^{-5}$ | 0.02 | $6.85e^{-6}$ |
| Protein | $2.09e^{-5}$ | $1.47e^{-5}$ | $1.41e^{-5}$ | $1.39e^{-5}$ | $1.38e^{-5}$ | 0.13 | 158.82 |
| ogbg-molbbbp | $3.94e^{-5}$ | $2.82e^{-5}$ | $2.74e^{-5}$ | $2.61e^{-5}$ | $2.54e^{-5}$ | $7.84e^{-4}$ | $1.40e^{-4}$ |

## 8.8 Statistics-based comparison with benchmark GGMs

Table 7 shows the benchmark results of statistics-based evaluation. On synthetic graphs, the GraphVAE-MM scores are superior to or competitive with the BiGG and GRAN scores. On the real-world graphs, the GraphVAE-MM scores are competitive with the BiGG and GRAN scores, and superior to the other benchmarks.

Table 7: Statistics-based comparison with benchmark GGMs. For a named evaluation graph statistic, each column reports the MMD between the test graphs and the generated graphs. The best result is in bold and the second best is underlined.

(a) Synthetic Graphs

| Method | Triangle Grid | | | | | Lobster | | | | | Grid | | | | |
|---|---|---|---|---|---|---|---|---|---|---|---|---|---|---|---|
| | Deg. | Clus. | Orbit | Spect | Diam. | Deg. | Clus. | Orbit | Spect | Diam. | Deg. | Clus. | Orbit | Spect | Diam. |
| 50/50 split | $3e^{-5}$ | 0.002 | $8e^{-5}$ | 0.004 | 0.014 | 0.002 | 0 | 0.002 | 0.005 | 0.032 | $1e^{-5}$ | 0 | $2e^{-5}$ | 0.004 | 0.014 |
| GraphVAE-MM | **0.001** | **0.093** | **0.001** | **0.013** | **0.133** | $2e^{-4}$ | **0** | _0.008_ | _0.017_ | _0.187_ | **$5e^{-4}$** | **0** | **0.001** | _0.014_ | **0.065** |
| GraphRNN-S [47] | 0.053 | 1.094 | 0.121 | 0.033 | 1.124 | 0.016 | 0.319 | 0.285 | 0.045 | 0.242 | 0.014 | 0.003 | 0.090 | 0.112 | _0.128_ |
| GraphRNN [47] | _0.033_ | 1.167 | 0.107 | 0.030 | _1.121_ | 0.004 | 0 | 0.033 | 0.035 | 0.384 | 0.013 | 0.166 | 0.019 | 0.111 | 0.460 |
| GRAN [32] | 0.134 | 0.678 | 0.673 | 0.184 | 1.133 | 0.005 | _0.304_ | 0.331 | 0.043 | 0.446 | 0.003 | $1e^{-4}$ | 0.007 | **0.012** | 0.281 |
| BiGG [11] | **0.001** | _0.107_ | _0.004_ | _0.020_ | 1.123 | **0.001** | **0** | **$6e^{-4}$** | **0.012** | **0.101** | 0.002 | _$3e^{-5}$_ | _0.003_ | 0.018 | 0.328 |

(b) Real Graphs

| Method | Protein | | | | | ogbg-molbbbp | | | | |
|---|---|---|---|---|---|---|---|---|---|---|
| | Deg. | Clus. | Orbit | Spect | Diam. | Deg. | Clus. | Orbit | Spect | Diam. |
| 50/50 split | $4e^{-5}$ | 0.004 | $5e^{-4}$ | $4e^{-4}$ | 0.003 | $2e^{-4}$ | $2e^{-5}$ | $9e^{-5}$ | $5e^{-4}$ | 0.002 |
| GraphVAE-MM | _0.006_ | **0.059** | _0.152_ | _0.007_ | _0.091_ | **0.001** | 0.005 | $8e^{-4}$ | **0.005** | **0.018** |
| GraphRNN-S [47] | 0.046 | 0.324 | 0.316 | 0.028 | 0.302 | 0.016 | 0.572 | 0.006 | 0.045 | 0.199 |
| GraphRNN [47] | 0.012 | 0.123 | 0.264 | 0.018 | 0.342 | _0.002_ | **$9e^{-4}$** | _$4e^{-4}$_ | 0.135 | 0.495 |
| GRAN [32] | **0.003** | **0.059** | **0.053** | **0.004** | **0.009** | 0.008 | 0.353 | 0.013 | 0.056 | 0.317 |
| BiGG [11] | 0.007 | _0.099_ | 0.316 | 0.012 | 0.181 | 0.003 | _0.001_ | **$5e^{-5}$** | _0.007_ | _0.033_ |

## 8.9 Comparison of GGMs on train and generation time

Table 8 compares benchmark GGMs and VGAEs on generation and train time for the main datasets. The benchmark GGMs require substantially more generation time than GraphVAEs. While MM modeling slows down training for the GraphVAE, the training time is still less than the benchmarks.

Table 8: Comparison of benchmark GGMs on the *train and generation time*. Train and Generation show average training time per epoch and average generation time per batch respectively. The best result is in bold and the second best is underlined.

| Method | Grid | | Lobster | | Triangle Grid | | Protein | | ogbg-molbbbp | |
|---|---|---|---|---|---|---|---|---|---|---|
| | Train (s) | Generation (s) | Train (s) | Generation (s) | Train (s) | Generation (s) | Train (s) | Generation (s) | Train (s) | Generation (s) |
| GraphVAE | **0.28** | **0.00** | **0.11** | **0.00** | **0.24** | **0.00** | **2.36** | **0.00** | 1.11 | **0.00** |
| GraphVAE-MM | 0.49 | **0.00** | 0.15 | **0.00** | 0.40 | **0.00** | 4.87 | **0.00** | 2.70 | **0.00** |
| GraphRNN-S [47] | 10.02 | 32.20 | 4.09 | 24.21 | 82.32 | 12.62 | 369.44 | 110.23 | 2.40 | 37.92 |
| GraphRNN [47] | 16.16 | 294.53 | 4.66 | 72.68 | 296.76 | 16.33 | 864.1 | 236.24 | 2.02 | 30.89 |
| GRAN [32] | 7.62 | 22.13 | 1.12 | 1.34 | 12.61 | 29.27 | 9.51 | 44.19 | 4.62 | 37.60 |
| BiGG [11] | 97.75 | 2.00 | 18.02 | 0.31 | 82.86 | 2.59 | 130.28 | 2.93 | 23.14 | 0.11 |

## 8.10 Details on extended experiments

This section extends the experiments on real graph sets by evaluating the micro-macro modeling on MUTAG, PTC, and QM9 datasets.

- MUTAG is a dataset of 188 mutagenic aromatic and heteroaromatic nitro compounds [46].

- PTC is a dataset of 344 chemical compounds that reports the carcinogenicity of male and female rats [46].

- QM9 is a large dataset containing about 134k organic molecules [39].

Following [48] we withhold the edges/nodes labels to obtain homogenized datasets in which two nodes are connected if there is an edge between them in the original graph. Table 9 shows statistics-based and GNN-based evaluation of GraphVAE with and without micro-macro modeling and comparison of micro-macro modeling on the train and generation time. Table 10 compares GraphVAE–MM with benchmark GGMs. It was not feasible to train all auto-regressive benchmarks on QM9, so we report results only for the GraphVAE architecture with and without micro-macro modeling. GraphVAE and GraphVAE-MM are trained for $20,000$ epochs except for QM9, a large dataset with more than $134k$ graphs, which are trained for $100$ epochs. The experiments follow the setting in the main body of the paper, for details see section 8.4.

**Impact on GraphVAE.** Table 9 shows a very large improvement from micro-macro modeling for MMD RBF of generated graphs, especially for PTC the MMD RBF is reduced by more than one order of magnitude. The diversity of generated graphs, F1 PR, also substantially increased, The magnitude of the increase is up to 53%. For QM9, also the reality of generated graphs slightly improved. GraphVAE already had a strong performance on QM9, compared to the other datasets. To calculate the ideal score, we used two randomly selected subsets of QM9 dataset, each including $5k$ graphs, because statistic-based evaluation metrics are slow to compute [44].

**GraphVAE-MM vs. Benchmark GGMs.** As table 10a shows, GraphVAE-MM beats the baselines on the MMD RBF, and is very competitive on the F1 PR, though worse than BiGG. In addition, in our experiments, GraphVAE-MM is much faster than the auto-regressive baselines in generation time, see 10c.

## 8.11 System architecture

The code for all models is run on the same system, an Intel(R) Core(TM) i9-9820X CPU 3.30GHz and Nvidia TITAN RTX GPU with TU102-core. Because of package compatibility issues, GraphRNN(-S) is run on an Intel(R) Core(TM) i7-5820K CPU 3.30GHz and a GM200 GeForce GTX TITAN X.

Table 9: Evaluation of micro-macro modeling for GraphVAEs on MUTAG, PTC, and QM9 datasets.

(a) GNN-based evaluation of micro-macro modeling.

| Method | MUTAG | | PTC | | QM9 | |
|---|---|---|---|---|---|---|
| | MMD RBF | F1 PR | MMD RBF | F1 PR | MMD RBF | F1 PR |
| 50/50 split | $0.03 \pm 0.00$ | $98.58 \pm 0.00$ | $0.04 \pm 0.00$ | $98.58 \pm 0.00$ | $0.010 \pm 0.00$ | $99.90 \pm 0.20$ |
| GraphVAE | $0.09 \pm 0.02$ | $78.38 \pm 10.50$ | $0.53 \pm 0.13$ | $31.96 \pm 16.00$ | $0.024 \pm 0.01$ | $\mathbf{97.28 \pm 0.03}$ |
| GraphVAE-MM | $\mathbf{0.07 \pm 0.01}$ | $\mathbf{86.63 \pm 10.59}$ | $\mathbf{0.04 \pm 0.01}$ | $\mathbf{84.40 \pm 5.60}$ | $\mathbf{0.019 \pm 0.00}$ | $96.16 \pm 0.01$ |

(b) Statistics-based evaluation of micro-macro modeling.

| Method | MUTAG | | | | | PTC | | | | | QM9 | | | | |
|---|---|---|---|---|---|---|---|---|---|---|---|---|---|---|---|
| | Deg. | Clus. | Orbit | Spect | Diam. | Deg. | Clus. | Orbit | Spect | Diam. | Deg. | Clus. | Orbit | Spect | Diam. |
| 50/50 split | $3e^{-4}$ | 0 | $1e^{-5}$ | 0.005 | 0.013 | $1e^{-4}$ | $9e^{-5}$ | $8e^{-5}$ | 0.002 | 0.013 | $5e^{-5}$ | $4e^{-4}$ | $4e^{-4}$ | $7e^{-5}$ | $2e^{-6}$ |
| GraphVAE | 0.005 | 0.126 | 0.003 | **0.019** | 0.055 | 0.197 | 0.757 | 0.562 | 0.036 | 0.143 | 0.007 | **0.002** | **0.003** | 0.004 | 0.005 |
| GraphVAE-MM | **0.001** | **0** | $\mathbf{1e^{-4}}$ | **0.019** | **0.015** | **0.020** | $\mathbf{3e^{-4}}$ | **0.003** | **0.018** | **0.043** | **0.005** | **0.002** | **0.003** | **0.003** | **0.004** |

(c) Comparison of micro-macro modeling on the train and generation time *per-epoch* and *per-batch* respectively

| Method | MUTAG | | PTC | | QM9 | |
|---|---|---|---|---|---|---|
| | Train (s) | Generation (s) | Train (s) | Generation (s) | Train (s) | Generation (s) |
| GraphVAE | **0.07** | $\mathbf{3e^{-4}}$ | **0.23** | $\mathbf{5e^{-4}}$ | **0.38** | $\mathbf{3e^{-4}}$ |
| GraphVAE-MM | 0.15 | $\mathbf{3e^{-4}}$ | 0.32 | $\mathbf{5e^{-4}}$ | 0.75 | $\mathbf{3e^{-4}}$ |

## 8.12 Complexity bounds

Table 11 gives complexity bounds for the different default statistics we use in this paper. Table 12 presents computational complexity of GGMs. The bounds are based on the literature and the analysis presented in this paper.

## 8.13 Societal impact

Graph generation could have both positive and negative Societal impact, depending on the application domain. On the positive side, graphs can represent molecules, and graph modeling supports medical discovery. On the negative side, network analysis ,as a field, has potential to increase and misuse control over network participants. For example, to motivate surveillance violations of privacy in targeting recommendations, or identify users through their social links. However, these harms can be mitigated by strengthening privacy protections during data collection. Furthermore, network analysis can provide significant societal benefit, for example, by highlighting the existence and situation of marginalized communities and understanding the flow of influence and (mis)information in social networks. The main contribution of our work supports more beneficial uses by enhancing the understanding of global network structure, as opposed to surveillance and the potentially harmful targeting of individuals.

## 8.14 Code overview

The implementation is provided at `https://github.com/kiarashza/GraphVAE-MM`. main.py includes the training pipeline and also micro-macro objective functions implementation. Source codes for loading real graph datasets and generating synthetic graphs are included in data.py. All the Python packages used in our experiments are provided in environment.yml. Generated graph samples for each of the datasets are provided in the "ReportedResult/" directory, both in the pickle and png format. This directory also includes the log files and hyperparameters details used to train the GraphVAE-MM on each of the datasets.

Table 10: Micro-macro modeling comparison with benchmark GGMs on MUTAG, and PTC datasets. The benchmark methods were infeasible on QM9. The best result is in bold and the second best is underlined.

(a) GNN-based comparison of micro-macro modeling with benchmark GGMs.

| Method | MUTAG | | PTC | |
|---|---|---|---|---|
| | MMD RBF | F1 PR | MMD RBF | F1 PR |
| 50/50 split | $0.03 \pm 0.00$ | $98.58 \pm 0.00$ | $0.04 \pm 0.00$ | $98.58 \pm 0.00$ |
| GraphVAE-MM | $\mathbf{0.07 \pm 0.01}$ | $86.63 \pm 10.59$ | $\mathbf{0.04 \pm 0.01}$ | $\underline{84.40 \pm 5.60}$ |
| GraphRNN-S [47] | $0.83 \pm 0.14$ | $55.25 \pm 22.62$ | $0.66 \pm 0.15$ | $34.50 \pm 18.12$ |
| GraphRNN [47] | $1.64 \pm 0.05$ | $0.99 \pm 0.00$ | $0.88 \pm 0.15$ | $32.26 \pm 0.05$ |
| GRAN [32] | $\underline{0.29 \pm 0.08}$ | $\underline{93.24 \pm 3.68}$ | $0.17 \pm 0.02$ | $81.20 \pm 7.14$ |
| BiGG [11] | $0.56 \pm 0.00$ | $\mathbf{98.08 \pm 0.00}$ | $0.04 \pm 0.00$ | $98.11 \pm 1.66$ |

(b) Statistics-based comparison of micro-macro modeling with benchmark GGMs.

| Method | MUTAG | | | | | PTC | | | | |
|---|---|---|---|---|---|---|---|---|---|---|
| | Deg. | Clus. | Orbit | Spect | Diam. | Deg. | Clus. | Orbit | Spect | Diam. |
| 50/50 split | $3e^{-4}$ | 0 | $1e^{-5}$ | 0.005 | 0.013 | $1e^{-4}$ | $9e^{-5}$ | $8e^{-5}$ | 0.002 | 0.013 |
| GraphVAE-MM | $\underline{0.001}$ | **0** | **1e⁻⁴** | **0.019** | **0.015** | 0.020 | **3e⁻⁴** | 0.003 | $\underline{0.018}$ | $\underline{0.043}$ |
| GraphRNN-S [47] | 0.006 | $\underline{5e^{-4}}$ | 0.002 | 0.105 | 1.157 | 0.022 | 0.254 | 0.035 | 0.057 | 0.270 |
| GraphRNN [47] | 0.006 | 0.210 | $\underline{8e^{-4}}$ | 0.070 | 0.819 | $\underline{0.005}$ | 0.003 | $\underline{0.002}$ | 0.075 | 0.397 |
| GRAN [32] | $\mathbf{6e^{-4}}$ | 0.015 | 0.007 | 0.053 | 0.685 | 0.013 | 0.137 | 0.006 | 0.034 | 0.194 |
| BiGG [11] | 0.004 | **0** | 0.002 | $\underline{0.040}$ | $\underline{0.293}$ | **1e⁻⁴** | $\underline{0.002}$ | $3e^{-5}$ | **0.016** | **0.015** |

(c) Comparison of micro-macro modeling on the train and generation time, *per-epoch* and *per-batch* respectively, with benchmark GGMs.

| Method | MUTAG | | PTC | |
|---|---|---|---|---|
| | Train (s) | Generation (s) | Train (s) | Generation (s) |
| GraphVAE-MM | **0.15** | **3e⁻⁴** | **0.32** | **5e⁻⁴** |
| GraphRNN-S [47] | 1.18 | 5.77 | 2.12 | 21.62 |
| GraphRNN [47] | 1.38 | 5.97 | 2.08 | 26.15 |
| GRAN [32] | $\underline{0.88}$ | 24.63 | $\underline{0.61}$ | 35.58 |
| BiGG [11] | 5.20 | $\underline{0.08}$ | 7.66 | $\underline{0.07}$ |

Table 11: Complexity of MM-ELBO Components

| Component | Time Complexity | Space Complexity | Property |
|---|---|---|---|
| Edge Reconstruction Probability | $O(N^2)$ | $O(N^2)$ | Permutation Equivariant |
| Triangle Count | $O(N^3)$ | $O(N^2)$ | Permutation Invariant |
| Degree histogram | $O(N^2)$ | $O(N^2)$ | Permutation Invariant |
| S-Step transition probability | $O(N^3)$ | $O(N^2)$ | Permutation Equivariant |

Table 12: Graph Generative Models complexity comparison

| Model | Train (Computational Complexity) | Graph Generation (Computational Complexity) | Auto-Regressive Decision Steps |
|---|---|---|---|
| GraphVAE [43] | $O(N^4)$ | $O(N^2)$ | $O(1)$ |
| GraphVAE | $O(N^2)$ | $O(N^2)$ | $O(1)$ |
| GraphVAE-MM | $O(N^3)$ | $O(N^2)$ | $O(1)$ |
| BiGG [11] | $O(min((|E| + N) \log N, \ N^2))$ | $O(min((|E| + N) \log N, \ N^2))$ | $\log N$ |
| GRAN [32] | $O(N^2)$ | $O(N^2)$ | $N$ |
| GraphRNN [47] | $O(N^2)$ | $O(N^2)$ | $|E|N$ |