# OpenReview forum: "Micro and Macro Level Graph Modeling for Graph Variational Auto-Encoders"
_NeurIPS.cc/2022/Conference — NeurIPS 2022 Accept_

### Official Review · Reviewer_dtRG · 2022-07-04

**Rating:** 6
**Confidence:** 3
**Soundness:** 3 good
**Presentation:** 2 fair
**Contribution:** 2 fair

**Summary:**

The authors of this paper newly presented a function that can reflect graph statistics in the graph generative model. They have shown various experiments and visualizations proving graph statistics are well-reflected. In addition, designing an objective function to reflect different graph statistics simply is a significant contribution.

**Questions:**

1. As a result of examining the paper or code, it seems that node feature or edge feature is not generated. In the case of using the Real-world dataset, it is considered important to generate not only the structure of the graph but also the node/edge feature well. Can you suggest an experiment related to this part?

2. As described in the paper, graph statistics are reflected using the calibrated Gaussian framework. There are two questions about this.
        a) Are graph statistics used for performance evaluation or reflected in objective function actually following Gaussian distribution? For example, if you think about 'degrees', it seems that the skew distribution is easy to appear due to the nature of the graph dataset. If so, it seems necessary to verify whether 'degrees' can be assumed to follow Gaussian distribution.
        b) Can graph statistics be reflected in other ways without utilizing the calibrated Gaussian framework? Is there an experiment that can support the evidence using the Calibrated Gaussian framework?

3. Figure 1 shows 'connected components' as an example to be improved in this paper. It would be nice to show that the proposed model has improved this part well.


**Ethics Review Area:**

["I don’t know"]

**Limitations:**

The positive social impacts presented in this paper include molecular presentation and medical discovery. However, since the model proposed by real-world dataset shows weak performance, it is seen as an important limitation.

**Strengths And Weaknesses:**

Originality: (Yes)
The proposed method seems to be original in that the authors proposed a new but simple VAE-based objective function to reflect graph statistics.

Quality: (Neutral)
Since the purpose of this study is to generate graphs that reflect graph statistics, theoretical support and experiments for the purpose are well shown. However, the performance on real-world datasets such as the molecule is marginal. In particular, when only one graph statistic is used, the performance degradation is greater than that of GraphVAE, which needs clarification. Since it shows good performance only when all three statistics presented in the paper are written, it is necessary to explain why these three combinations were selected and what synergy they show.

Clarity: (Yes)
There was no difficulty in understanding what the paper was trying to say, and it shows sufficient proof of the formula. I think it would be easier to understand if the architecture overview was attached. I suggest adding a picture of the structure for the reader's clear understanding.

Significance: (Neutral)
This model seems to have particular strengths in experiments using Synthetic datasets. In addition, it seems to be a good contribution that it showed a higher performance improvement compared to GraphVAE. However, as discussed in the paper, performance in real-world datasets seems to be more important to contribute in practical areas such as molecule and medical discovery. However, the experimental results presented in the paper do not support this.  Additional experiments will be needed to show that graphs are well generated using the QM9 dataset shown in GraphVAE.

---

> ### Author Response · Authors · 2022-08-02
>
> **Limitations:**
> The positive social impacts presented in this paper include molecular presentation and medical discovery. However, since the model proposed by real-world dataset shows weak performance, it is seen as an important limitation.
>
> We are *puzzled* by this comment.  If the comparison is between GraphVAE with and without micro-macro modelling, Table 1 shows a big improvement, *especially on the real-worlds graphs*. If the comparison is between GraphVAE and auto-regressive baselines, GraphVAE-MM beats the baselines on the MMD RBF, graph fidelity (realism) metric, and is very competitive on the F1 PR, diversity metric. In addition, in our experiments GraphVAE-MM (as well as GraphVAE) are much faster than the auto-regressive baselines both in generation and train time (Figure 3 and table 9). This is true for all datasets, including the real-world ones. *The other reviews* seem to share our conclusions. Next we go through the details of our evidence, including some new results following your helpful suggestions.
>
> ### The improvement in the real-worlds datasets (Table 1)
>
> *Ogbg-molbbbp dataset.* Table 1 shows that the micro-macro modeling improvement on Ogbg-molbbbp is substantive and *bigger than* any other datasets, including *synthetic datasets*. The magnitude of the increase is 39.35 in F1 PR  and 0.18 reduction in MMD RBF. In percentages, the improvement is 72% and 111% in F1 PR and MMD respectively which indicate significant improvement in both fidelity (reality) and diversity of generated graphs and achieving a near-perfect score.
>
> *Protein dataset.* The GraphVAE-MMD RBF score improves by 70% and achieves 0.03 which is almost the ideal score, i.e. the MMD RBF of 50/50 split of the test set. Figure 10  in the appendix also supports our contention. The figure visually contrasts generated Protein graphs by GraphVAE and GraphVAE-MM, where GraphVAE-MM, matches the complicated patterns in the  target graphs the best.
>
> ###  Choice of real graph datasets
> The paper studies Protein and the ogbg-molbbbp real-world datasets. Previous studies have also used 2 real-world datasets [43, 32, 11]. We replaced the Cloud dataset from these studies with the ogbg-molbbbp because it was not feasible to train the GraphRNN baseline on the Cloud dataset.  Also the ogbg-molbbbp dataset is well known in the community from the Open Graph Benchmark [23]. Protein and ogbg-molbbbp are from biology with information about proteins and molecules respectively.
>
> ### New Results Following the Reviewer Suggestions
> We do agree that more experiments are always better, and especially experiments on real-world datasets. We had used the **QM9** dataset that you mentioned but were not able to train the auto-regressive baselines on it because they do not scale well in the number of graphs; this dataset has more than 130K graphs. The revision now shows the QM9 improvements from micro-modeling by comparing GraphVAE vs. GraphVAE-MM (Table 8 in the appendix).
> For QM9 the MMD RBF of generated graphs improved, In percentages, by 15%. Given the already strong performance of GraphVAE on QM9, this is a substantive improvement and provides evidence of Micro-macro modeling effectiveness. Thank you for the suggestion.
>
> We have added experiments on the real-world **MUTAG** and **PTC** benchmarks [44], and the performance of GraphVAE-MM has held up well. (See Tables 8 and 9 in Section 8.10 of the revised appendix, we also updated our anonymous [*GitHub*](https://github.com/ddccbbee/GraphVAE-MM)).  On MUTAG and PTC is the improvements from micro-modelling are even better than those we reported on the Protein and ogbg-molbbbp datasets. On MUTAG, substantive improvements over our strongest baseline, the BiGG auto-regressive method. On PTC, the pattern is similar to those reported in the main body of the paper: Substantive improvement in generation quality over all baseline except for BiGG, competitive quality and much faster runtime and generation than the autoregressive baselines.

---

> ### Author Response · Authors · 2022-08-02
>
> **Q2.** As described in the paper, graph statistics are reflected using the calibrated Gaussian framework. There are two questions about this.
>
> **a)** Are graph statistics used for performance evaluation or reflected in objective function actually following Gaussian distribution? For example, if you think about 'degrees', it seems that the skew distribution is easy to appear due to the nature of the graph dataset. If so, it seems necessary to verify whether 'degrees' can be assumed to follow Gaussian distribution.
>
> Thanks for the question. We agree with the reviewer that graph statistics may have non-gaussian distribution. However, we should clarify that the gaussian distribution is used for the conditional distribution $p(F_u|z)$ (probability of observed graph statistic given the graph embedding). General VAE theory indicates that the marginal distribution over graph statistics,$\int p(F_u|z)p(z)dz$, can in principle fit any distribution over graph statistics, including skewed ones. Empirically, the MMD metrics are showing a close match between the observed distribution over graph statistics and the distribution implicitly defined by our trained GraphVAE-MM model, including node degree. Next, we expand on this point in some detail.
>
> VAE can capture high-dimensional complicated data distributions, and it is widely applied to various data, such as images, videos, and audio and speech. With respect to graph statistics (as opposed to edges), our model behaves like a VAE [26]. The VAE decoder outputs a Gaussian distribution over statistics/feature vectors given a latent variable z. This does not mean that the statistic/feature vector has an unconditional or marginal Gaussian distribution. The data distribution $p_\theta(X)$ is given by the unconditional/marginal distribution $\int p(X|z)p(z)dz$. The VAE can in principle (with a powerful enough decoder) model any input data distribution. Similar to VAE, GraphVAE-MM can accommodate any distribution over graph statistics with a sufficiently powerful decoder.
>
>
> **b)** Can graph statistics be reflected in other ways without utilizing the calibrated Gaussian framework? Is there an experiment that can support the evidence using the Calibrated Gaussian framework?
>
> We could use a Gaussian VAE framework in the decoder where the variances are treated as hyperparameters (as in the original VAE paper). We obtained good empirical results with hyperparameter search, but it is a time-consuming process and we believe would deter users from applying our method. Adapting the Calibrated Gaussian (with a novel standardization for graph statistics with divergent scales) allows us to avoid hyperparameter search without a loss of performance.
> It would be fairly simple to condition on graph statistics by using them as part of the input to the encoder (e.g. using the Graph NNN framework [deepmind]). The difficulty is to generate graph statistics by adding to the decoder the ability to output them. Since we are using a graphVAE to generate adjacencies, it is natural to use a VAE to generate graph statistics as well.
>
> We believe that our results about the usefulness of adding graph statistics to a generative model will stimulate future research with other generative frameworks to answer your question. Modelling graphs at two different scales opens a new line of research. In section 6 we discuss micro-macro modelling for other GGM architectures [lines 310-317]. For example for GANs we suggest that “A way to combine MM modeling with GANs is to augment the input to the discriminator with graph statistics computed for both real and generated graphs.” In this approach there is no explicit probabilistic model over graph statistics, hence no conditional (calibrated) Gaussian.
>
> **Q3.** Figure 1 shows 'connected components' as an example to be improved in this paper. It would be nice to show that the proposed model has improved this part well.
>
> Thanks for the suggestion. In our experiments we reported graph diameter, a new evaluation metric not previously used, which depends strongly on graph connectivity.
> Empirical results for the graph diameter, as illustrated in tables 2 and 8.b, shows significant improvement in the MMD between the test graphs and the generated graphs diagram by applying the micro-macro modeling of GraphVAE-MM. The improvement ranges  up to 2 orders of magnitude improvement in 7 out of 8 studied datasets.
>
>
> **Comment 2.** I think it would be easier to understand if the architecture overview was attached. I suggest adding a picture of the structure for the reader's clear understanding.
>
> Thanks for the suggestion, the figure is added to the paper, see Appendix figure 4.b in the revised version.

---

> ### Author Response · Authors · 2022-08-02
>
> We thank the reviewer for the care and attention devoted to the paper. Below our responses to the questions and comments can be found. Please note that all references and citations refer to the paper.
>
> **Q1.** As a result of examining the paper or code, it seems that node feature or edge feature is not generated. In the case of using the Real-world dataset, it is considered important to generate not only the structure of the graph but also the node/edge feature well. Can you suggest an experiment related to this part?
>
>
> The model can simply expand to generate the node and edge features. For example, as in the GraphVAE [41], the decoder generates the node feature, X, and edge features, E. The micro-macro (MM) loss would be of the form:
>
> $L_θ (A, E, X) = L^{0}_{θ} (A) + γ L^1_θ (F_1, . . . , F_m) + ΓL^2_θ (X,E)$
>
> where $Γ$ is a hyperparameter that balances the feature aspect.
>
> We agree that attributed graphs are an important topic. As you point out, a strength of the GraphVAE [42] approach is that it accommodates node and edge features. However, this is not true of the auto-regressive baselines [43, 32, 11] and other traditional graph models (going back to Erd ̋os and Rén) that focus on the problem of learning structural information from graph data. As the auto-regressive baselines are considered SOTA by many (e.g Hamilton [20]), we wanted to use exactly their setting. In our view, the ease with which GraphVAEs [41] can accommodate node/edge features is another advantage over current auto-regressive methods (in addition to the other GraphVAE [41] strengths we demonstrate in the paper).
> Our paper follows graph structure learning research track and shows that jointly modeling local and global level properties can hugely affect learning the structure of graph data. We have added as a future direction evaluating the impact of micro-macro modeling on graphs with node or edge features, thank you for the suggestion (lines 335-337).

---

> ### Comment · Reviewer_dtRG · 2022-08-09
> **Response to rebuttal**
>
> Thank you for your answers on my questions and your additional experiments on our suggestions. Lots of my doubtful points have been fulfilled.
> However, when I look thoroughly at the results of target statistics ‘Degree’, ‘Cluster’, and ‘orbit’ of ‘ogbg-molbbbp’ dataset in Table 5 (page 16, section 8.7.), it can be seen that the performance of your model got worse if you use only one of the graph statistics, while your model showed a good performance if you use all of three suggested statistics together.
>
> I have the following two questions regarding this observation.
> 1. Can you explain why your model performs poorly when you use only one graph statistic?
> 2. I wonder why the performance improved rapidly when you use all of three statistics. Does this result imply that using more and more graph statistics brings better performance, or combination of those three statistics you used shows specifically good performance?
>
> Overall, I really appreciate your detailed and faithful responses.
> I would change my rating to Weak Accept, 6.

---

> > ### Author Response · Authors · 2022-08-09
> >
> > We thank the reviewer for re-evaluating our work and increasing their rating.  Below are responses to the follow-up questions.
> >
> > **Q.** Can you explain why your model performs poorly when you use only one graph statistic? I wonder why the performance improved rapidly when you use all of three statistics. Does this result imply that using more and more graph statistics brings better performance, or combination of those three statistics you used shows specifically good performance?
> >
> >
> > As the second block of table 5 shows, and as expected, different statistics are more important for different datasets and have different effects. However, as mentioned in lines 574-576 no single graph statistic has the power of all three combined.
> >
> > The framework uses a combination of the three graph statistics as default and the empirical experiment shows the improvement ranges up to 2 orders of magnitude improvement in 8 studied datasets with different properties. However as mentioned in lines  168 and 169, the default graph statistics can be extended for specific target statistics. Our experimental result shows that, depending on the dataset, extending the default statistics by adding more graph statistics generally results in better performance, however, it arises additional computational overhead.
> >
> > We agree there is an interesting research question around the interaction of different graph statistics that are opened up by the framework.
> >  In Section 7, Conclusion and future work [lines 331-335], we mention that “micro-macro modeling opens a number of fruitful avenues for future work. i) Investigating which graph statistics are important for generating which types of graphs. This connects with the rich area of graph kernels [34] that are often based on graph statistics. ii) Investigating which graph statistics are important for particular domains.”

---

### Official Review · Reviewer_Uc7B · 2022-07-11

**Rating:** 6
**Confidence:** 4
**Soundness:** 3 good
**Presentation:** 3 good
**Contribution:** 3 good

**Summary:**

This paper jointly models micro and macro level graph information for graph generation. A principled joint probabilistic model for both levels is proposed and an ELBO training objective is derived for graph encoder-decoder models. Extensive experiments and visualization results validate the efficacy of adding micro-macro modelling to GraphVAE models for graph generation.

**Questions:**

The proposed MM objective function is applied on GraphVAE for an AB design and its effectiveness is showed for graph generation. Do you think if the benefits of micro-macro modeling would generalise to other models or other graph tasks? Some related discussions in this regard are necessary.

**Limitations:**

The authors have adequately discussed the limitations of their work.

**Strengths And Weaknesses:**

Strengths:
1. This paper is well motivated and the idea of utilizing node-level properties and graph-level statistics to constrain graph generation seems reasonable.
2. The design of micro-macro (MM) loss is clear and theoretically solid.
3. The authors have done a thorough analysis of the proposed model and validated its effectiveness through qualitative and quantitative evaluation. The main claims are supported by the experimental results.

Weaknesses:
My main concern is that the proposed objective function is only applied on GraphVAE following an AB design. Although the experimental results are satisfactory on graph generation, it remains unclear whether the benefits of micro-macro modeling would generalise to other models.

---

> ### Author Response · Authors · 2022-08-02
>
> We thank the reviewer for the care and attention devoted to the paper. Below our responses to the questions can be found. Please note that all references and citations refer to the paper.
>
> **Questions:** The proposed MM objective function is applied on GraphVAE for an AB design and its effectiveness is shown for graph generation. Do you think if the benefits of micro-macro modeling would generalize to other models or other graph tasks? Some related discussions in this regard are necessary.
>
> We thank the reviewer for the attention devoted to the paper.  Below you will find our responses to the question.
>
> **Part 1.** Do you think if the benefits of micro-macro modeling would generalize to other *models*?
>
> Thanks for the interesting question. The proposed model is a new multi-scale perspective on graph modelling, not a new GGN architecture. For evaluation we choose a specific GraphVAE. In fact, we expect part of the impact of our paper will be to stimulate research into using MM modelling to improve the performance of many graph architectures including Autoregressive and GANs (see section 6, lines 310-317, also future work, line 334.)
>
>
> **Part 2.** Do you think if the benefits of micro-macro modeling would generalize to other graph *tasks*?
>
> We thank the reviewer for bringing up this interesting research question. We believe exploiting graph statistics can potentially be utilized in many studies of deep graph-level representation learning. Since the encoder of our GraphVAE-MM models is trained to produce a graph embedding, natural downstream tasks are graph classification, graph clustering, and visualization. We have started experiments on graph classification using GraphVAE-MM and are observing improvements. This paper focuses on graph generation, one of the research frontiers of graph representation learning.
>
> We have discussed and added the possibility of using graph-level statistics in unsupervised graph-level representation learning and its effect on downstream tasks as future work, see lines 335-337 in revised version.

---

### Official Review · Reviewer_KhAw · 2022-07-12

**Rating:** 6
**Confidence:** 3
**Soundness:** 3 good
**Presentation:** 3 good
**Contribution:** 3 good

**Summary:**

The contributions of this paper was to model graph data jointly at two levels: a micro level based on local
information and a macro level based on aggregate graph statistics.

**Questions:**

1. How the proposed model is different from the GAN-based model?

**Ethics Review Area:**

["I don’t know"]

**Limitations:**

Yes

**Strengths And Weaknesses:**

Positives:

1. The idea of this work is interesting and novel, which trys to use probabilistic model to explore the local and global graph statistics.

2. The performance of this work is very good, comparing to the existing GraphVAE. And the code is available.

Negathive:
1. The scalability of this work may be a challenge, the compexity of the descriptors is either O(N^2) or  O(N^3). Also the algorithm reuqires pre-define a graph descriptors to compute the graph statistics.

2. The algorithm part is straightforwad. Basically, it designs a MM loss into one unified fraemwork. It seems that many GAN-based models can achieve the similar function. Any discussion?

---

> ### Author Response · Authors · 2022-08-02
>
> We thank the reviewer for the care and attention devoted to the paper. Below our responses to the question can be found. Please note that all references and citations refer to the paper.
>
> **Questions**. How the proposed model is different from the GAN-based model?
>
>
> Thanks for the interesting question.
>
> In section 6 under “Micro-Macro modeling for other GGM architectures” we discuss other GGM architectures. The general difference to a GraphVAE architecture is that the GAN does not have an encoder component but adds a discriminator. We expand on our discussion of GANs specifically.
>
> To the best of our knowledge, GAN-based GGMs either 1) directly work on the adjacency matrix [8] or 2) mimic the graph by generating the random walks [7].
> Approach 1) adapts GAN and operates directly on graph adjacency matrices. The approach is a likelihood-free generative model in which the generator maps a graph latent to an adjacency matrix, and the discriminator classifies the adjacency matrix as real or synthetic.
> In approach 2), graphs are represented by generated random walks. The insight behind this approach is that transition counts can capture graph structure. NetGan and MolGan [7,8] are one of the popular GAN-based generative models which are mentioned in section 6 and adopt these approaches.
>
> As discussed in the paper [lines 21-22] and  in [9] both random walk and adjacency matrices are graph local level information which means the GAN models are limited to the local aspect of the graph, rather than using global graph-level statistics. The proposed micro-macro model is a new multi-scale perspective on graph modelling. In fact, we expect part of the impact of our paper will be to stimulate research into using MM modelling to improve the performance of many graph architectures including Autoregressive and GANs (see Section 6).
>
> For empirical comparison, we compared the NetGAN method with the proposed model, GraphVAE-MM, in statistics-based evaluation (Section 5.3 line 256). The table below compares NetGAN and the GraphVAE-MM on lobster and grid graph generation tasks. As shown, the proposed approach MM model graph structure metrics by 1-4 orders of magnitude compared to NetGAN.
>
> ---
>                 Dataset:                        Lobster                         Grid
> ---
>                 Stat:          Deg.     Clus.     Orbit     Spec        Deg.     Clus.     Orbit     Spec
> ---
>                 NetGAN [7]     1.56      0.03     0.86     3.20         1.97     1.31      0.95     0.46
> ---
>                 GraphVAE-MM    2e-4       0       0.008    0.017        5e-4     0        0.001     0.014
>
> Our experimental design closely followed that of recent SOTA papers in graph generation, which also did not compare with NetGAN.

---

### Official Review · Reviewer_BpMN · 2022-07-19

**Rating:** 4
**Confidence:** 4
**Soundness:** 3 good
**Presentation:** 3 good
**Contribution:** 2 fair

**Summary:**

This paper studies the problem of graph generation, and proposes a new model using both micro and macro level supervision information in GraphVAE architecture. Fitting adjacency matrix is the micro supervision, and three kinds of graph statistics, i.e., degree histogram, number of triangles, and higher-order proximity relations, are adopted as macro supervision. The object consists of ELBOs modeling micro-macro loss and a KL-divergence between the prior and the approximate posterior of hidden representation.  The proposed model is validated on 3 synthetic, and 2 real-world graph datasets. The experimental results the proposed model generates graphs with a lower discrepancy between generated and test graph embeddings than graphs generated by competitors in terms of MMD RBF and F1 PR.

**Questions:**

Q1.  How the graph generation task benefits from fitting graph statistics. And what kind of graph statistics should be chosen as targets？

Q2. How to form descriptor functions with respect to vector label histogram and triangle count?

**Limitations:**

Yes.

**Strengths And Weaknesses:**

Strong points:

S1. The macro objective of fitting graph statistics in graph generation is novel to me.

S2. The paper proposes a general micro-macro ELBO as the objective, and then implements the ELBO by graph neural networks.

S3. The experimental results show the proposed model outperforms the competitors.

Weak points:

W1. It is not clear how the graph generation task benefits from fitting graph statistics. In other words, what is the limitation to only fitting adjacency matrix in graph generation? From this line, I have a concern about what kind of graph statistics should be chosen as targets？
This paper selects three graph statistics, but does not present an explanation for this selection.

W2. The efficiency. Both calculating and fitting graph statistics bring new computing costs, e.g., the complexity is $O(n^3)$ to compute the transition probability matrix.

W3. It is not clear how to form descriptor functions with respect to vector label histogram and triangle count. And how to guarantee the descriptor functions are differentiable.

---

> ### Author Response · Authors · 2022-08-02
>
>
> **Q2.)** How to form descriptor functions with respect to vector label histogram and triangle count? It is not clear how to form descriptor functions with respect to vector label histogram and triangle count. And how to guarantee the descriptor functions are differentiable.
>
> Thanks for the question. Here we clarify the  definitions and explain how the descriptor functions differentiability is guaranteed.
>
> As mentioned in line 114, “a descriptor function is the function which maps an adjacency matrix A to a l-dimensional graph statistic”. Next we go over the default graph statistics and corresponding descriptor function which are used to calculate each of them.
>
> *Triangle count.* As explained in section 4, the number of triangles in a simple graph $A$ is a scalar and computed by
> $Tri(A) = \sum_i A^3_{ii}$    where $A$ is a soft adjacency matrix with $A_{ij}\in[0 \quad 1]$. Since matrix multiplication and summation is differentiable, so is the descriptor function, $Tri(A)$ with respect to $A_{ij}$.
>
> *S-Step transition probability kernel.* Similar to Triangle count, S-Step transition probability kernel is calculated by a simple (matrix) multiplication, $P^s(A) = {(D(A)^{−1}A)}^s$.  Since division, sum and matrix multiplication is differentiable $P^s(A)$, is also differentiable with respect to $A_{ij}$. Also see lines 186-191.
>
> *Vector Label Histogram (VLH).* As explained in lines 175 - 185, VLH is calculated by applying a soft histogram function to degree vector $V$, where $V_i =\sum_ j A_{ij}$.
> Learnable and differentiable histograms function have been used and studied in classification methods with differentiable end-to-end deep architectures, see  [Learnable Histogram: Statistical Context Features for Deep Neural Networks"](https://arxiv.org/abs/1804.09398)
>
>
>
> **Weakness W2.** Both calculating and fitting graph statistics bring new computing costs, e.g., the complexity is $O(N^3)$ to compute the transition probability matrix.
>
> The reviewer correctly pointed out the computational complexity of graph generation increases by adding graph statistics. One of the contributions of our paper is a set of techniques to achieve fast generation nevertheless (Table 7).  We discussed computational overhead from different aspects:
>
> *Training Overhead VS Generation Overhead.* The graph statistics only affect the model in the training and do not cause any overhead in the generation/test phase.
>
> *Computation Time.* As discussed in section 4, exploited graph statistics including transition matrix can be calculated in parallel in near constant time for small and medium size graphs.
>
> *GraphVAE-MM is substantially faster than GGM benchmarks.* Despite the computational overhead, compared to the popular benchmark GGMs, the GraphVAE-MM  has substantially less train and generation time, see figure 3 and table 7.
> Future Work. As discussed in section 6, line 307, Approximating graph statistics can reduce the computational cost significantly [24, 15, 31, 38] and can be exploited in future studies.
>
> We also note that edge reconstruction and evaluating the edge reconstruction probability are already expensive, and tends to dominate the graph statistic computations; see table 10.

---

> ### Author Response · Authors · 2022-08-02
>
> We thank the reviewer for the care and attention devoted to the paper. Below our responses to the comments and questions can be found. Please note that all references and citations refer to the paper.
>
> **Q1.a)**  How the graph generation task benefits from fitting graph statistics. In other words, what is the limitation to only fitting an adjacency matrix in graph generation?
>
> Thanks for the important question. The “Motivation” section in the introduction was meant to address the general benefits of fitting graph statistics. We briefly review and expand on our arguments there.
>
> *General Motivation.* We list two key properties: user control and graph realism (lines 30-33). For user control, a user may know which target graph statistics are important in their domain (See also the discussion in ref 33). In our MM framework, the user only needs to specify the target graph statistics and learning will automatically select graph models that match them (line 36-37). For example for a large payment graph recording economic transactions, a macro economist may be mainly interested in the average price level of a target goods basket. For a central bank managing a payment system, the total number of transactions and the maximum payment amount may be more important. Allowing the user to specify target statistics is a way to incorporate their domain knowledge.
>
> As for graph realism, compared to standard GGM objective functions that are based on predicting individual adjacencies, matching graph statistics serves as a regularizer for latent representations that increases the realism of the generated graph structures (lines 31-32). *This is because standard objective functions based on adjacency matrices alone weight all edges equally.* However, adjacencies (or non-adjacencies) have different roles in the graph global structure. Some edges play a critical role in maintaining the connectivity/community structure, while the rest are less important. Figure 1 in the paper illustrates this difference. Graph statistics can reflect the different roles. For example, consider K-step transition probability, which is utilized in the paper. The K-step transition probability matrix, k>1, encodes the connectivity information of the graph (see also discussion in lines 192-193).
>
> *Another issue with modelling adjacency matrices only is that they are sparse.* They are highly imbalance with most (>90%) entries being 0. By directly matching the adjacency matrix, a model tends to generate an overly sparse graph. To address the graph sparse structure Kipf and Welling [27] used weighted cross-entropy. However, it has been shown that weighted cross-entropy can result in distortion in measuring the quality of reconstructed data [38].
> On the other hand, graph global statistics are generally a scalar or dense matrix/vector of real values. For example, the number of triangles studied in this paper is a scalar, and as mentioned in line 191, K-steps transition probabilities are generally dense matrices. Our experiments show that the MM objective leads GraphVAEs to generate graphs with more realistic densities.
>
> *Empirical Improvement.* Our results show the benefits of adding graph statistics empirically. Specifically Tables 1, 2 and the extended experiment in the appendix following Reviewer dtRG’s suggestions, tables 8 and 9. Our experiments show that adding global properties to the GraphVAE, indicated as GraphVAE-MM, improves graph quality scores up to 2 orders of magnitude on eight benchmark datasets.
>
> **Q1.b)** And what kind of graph statistics should be chosen as targets？
>
>
> The paper uses three graph statistics as default statistics for applications where the user does not specify target statistics, and to evaluate the general idea of micro-macro modelling.
> As we discussed in section 4 [lines 170-202], our criteria for choosing graph statistics are as follows. 1) Meaningful and easy to interpret. 2) Differentiable with respect to the entries in a reconstructed soft adjacency matrix. 3) Permutation-equivariance. 4) Known from prior research to be generally important for graph modelling in different domains from prior research.
>
> We agree with the reviewer that investigating the statistics which should be chosen as targets is an interesting research question. In Section 7, Conclusion and future work [331-335], we mention that “micro-macro modeling opens a number of fruitful avenues for future work. i) Investigating which graph statistics are important for generating which types of graphs. This connects with the rich area of graph kernels [34] that are often based on graph statistics. ii) Investigating which graph statistics are important for particular domains.”

---

### Meta-Review · Area_Chair_JQrm · 2022-08-27

**Recommendation:** Accept
**Confidence:** Certain

**Metareview:**

This paper proposes a new generative model for the generation of graphs. Different from most of existing approaches, the proposed method considers both node and graph level properties to capture high-order connectivity and overcome sparsity of any observed graph. The writing is general clear and the results are convincing. The reviewers are overall positive, with some concerns on the motivation, which has been addressed well by the authors in the rebuttal. Some other questions raised by the reviewers are also appropriately addressed, which leads to the increase of some scores. The downside of the approach lies in the time complexity in collecting the macro-level statistics. But overall, it is a good paper worth accepting.

**Award:**

No

---

### Decision · Program_Chairs · 2022-09-14

Accept